# LASeR: Learning to Adaptively Select
# Reward Models with Multi-Armed Bandits

**Duy Nguyen**[1]*     **Archiki Prasad**[1]*     **Elias Stengel-Eskin**[1,2]     **Mohit Bansal**[1]
[1]UNC Chapel Hill      [2]The University of Texas at Austin

## Abstract

Reward Models (RMs) are crucial to aligning large language models (LLMs), but the degree to which an RM specialized to one task (e.g. writing) generalizes to new tasks (e.g. math) is often not known *a priori*, often making using only one fixed RM to train LLMs *suboptimal*. However, optimizing LLMs with multiple RMs simultaneously can incur a prohibitively high computational cost and lead to conflicting signals from different RMs that may degrade performance. To address these challenges, we introduce **LASeR** (**L**earning to **A**daptively **Se**lect **R**ewards), which frames reward model selection as a multi-armed bandit problem, efficiently and iteratively training LLMs using multiple RMs by selecting the most well-suited RM for each instance. On commonsense and math reasoning tasks, we show that LASeR boosts iterative LLM training, improving the absolute average accuracy of Llama-3-8B over three datasets by 2.67% over an ensemble of RM scores while also showing superior efficiency (e.g., a 2× speedup). Moreover, on WildChat (open-ended instruction-following tasks), LASeR leads to a 72.69% AlpacaEval win rate over the RM score ensemble baseline. Extending to long-context generation, LASeR improves by 2.96 F1 points (avg.) on single-document QA tasks and 2.97 F1 points on few-shot learning over the RM score ensemble baseline with best-of-$n$ sampling.[1]

## 1 Introduction

When comparing two responses, human preferences often differ depending on factors like the underlying task, who the annotators are [Santurkar et al., 2023, Ahmadian et al., 2024], and how preferences are elicited [Bansal et al., 2024]. Therefore, models of preference data are also likely to differ and might include noise as well as any biases contained in the preference data used to train them. This can pose a problem when using such models as "reward models" (RMs) to align large language models (LLMs) to human preferences using reinforcement learning with human feedback [Christiano et al., 2017, Ziegler et al., 2019, Ouyang et al., 2022]. Recent work has focused on aligning LLMs through iterative training, using reward models as proxies for human judgment [Gulcehre et al., 2023], leveraging the LLM to act as an implicit RM or judge [Yuan et al., 2024b, Chen et al., 2024b], or using the gold answer to compute a reward [Pang et al., 2024]. Under this paradigm, there are three stages to training LLMs: (i) *generating multiple responses* to a query from an LLM; (ii) *scoring the responses with an RM* to create preference data with better and worse responses; and (iii) using *model-generated preference data* to further train the LLM. Note that for most domains, the gold reward is not readily available, making the quality of the RM or the degree to which it reflects human preferences (i.e., the gold reward) crucial to improving LLM performance. Indeed, several prior efforts aim to train new RMs that better reflect human preferences [Lambert et al., 2024].

---

*Equal Contribution

[1]Code: https://github.com/duykhuongnguyen/LASeR-MAB

However, *selecting one reward model* to guide LLM training can be *suboptimal* for three main reasons: (1) A single RM may not generalize to heterogeneous sets of examples. RMs are typically designed to reflect specific objectives and may be trained on offline preference datasets. Thus, an RM that performs well on one dataset or domain *may not generalize effectively* to others, leading to misaligned outputs across different tasks or domains [Kirk et al., 2023, Chen et al., 2024a, Casper et al., 2023, Gao et al., 2023]. For instance, creativity plays a key role in evaluating the quality of a story, whereas correctness is more important in scoring math solutions. (2) RM performance leaderboards (e.g., Lambert et al. [2024]) that rely on human-annotated preferences can have unreliable rankings due to the presence of *incorrect and ambiguous preferences* [Yu et al., 2024, Hejna et al., 2023]. (3) Lastly, over-optimization on one RM can lead to *reward hacking* problem [Skalse et al., 2022, Rafailov et al., 2024a], even resulting in downstream performance drops.

To mitigate these issues, a prevalent approach is to *ensemble multiple reward models* [Coste et al., 2023, Eisenstein et al., 2023, Zhang et al., 2024, Ramé et al., 2024]. However, these methods also come with significant challenges: as RMs are typically based on LLMs, training with multiple RMs often requires loading and managing several large models simultaneously, which can be *computationally expensive*, becoming infeasible as models increase in size. Moreover, aggregating multiple RM scores together is susceptible to *noisy rewards or conflicting preferences* from RMs, especially RMs that are not well-suited for the specific task [Rita et al., 2024]. This, in turn, can degrade the quality of the preference data, leading to low-quality updates during training [Wang et al., 2024a]. Finally, manually selecting a subset of RMs to combine is a labor-intensive process that involves training many different variants on a combinatorially large set of RM groupings. This underscores the need for a more efficient and robust LLM training method with multiple RMs.

To fill this gap, we introduce **L**earning to **A**daptively **Se**lect **R**ewards (LASER), that, given a set of RMs, adaptively and efficiently *selects* a suitable RM for each instance by casting selection as a *multi-armed bandit* problem [Vermorel and Mohri, 2005, Audibert et al., 2009]. Specifically, during training, the RM (arm) is chosen dynamically based on contextual information about the model's performance and past interactions. The LLM is then fine-tuned based on the RM-annotated data, and the bandit's parameters are updated accordingly to reflect the performance of the LLM after training on preference data annotated using selected RM (see Fig. 1). By design, LASER's adaptive instance-level or batch-level RM selection (c.f. Sec. 3) addresses the three shortcomings of choosing one reward model: lack of generalization, unreliable rankings, and over-optimization. LASER eliminates the need for manual tuning or oracle RM selection, with users only tasked with selecting a pool of RMs from a leaderboard like RewardBench [Lambert et al., 2024] without needing to know in advance which RM is best suited for a specific task. Moreover, previous multi-RM methods do not explicitly analyze and address conflicting signals from multiple RMs, and require simultaneously loading and running multiple RMs [Ramé et al., 2024, Coste et al., 2023] or training an RM for a specific task [Quan, 2024]. On the other hand, LASER selects one RM at each training step (Sec. 4), avoiding conflicts between RMs and improving the overall training efficiency.

Empirically, we demonstrate the effectiveness of LASER for iteratively training LLMs using multiple RMs on three broad domains: reasoning, instruction-following in text generation, and long-context understanding (Sec. 4.2). We show that on reasoning benchmarks such as StrategyQA [Geva et al., 2021] (testing commonsense reasoning), GSM8K [Cobbe et al., 2021] (testing math reasoning), and MMLU [Hendrycks et al., 2021b] (testing general knowledge reasoning), LASER with Llama-3-8B improves absolute accuracy (averaged across 3 datasets) by $1.45\%$ over a baseline that uses best single RM for training and $2.67\%$ over an ensemble of RM scores baseline. LASER is also effective on general instruction-following: we show that using LASER with four strong 7B RMs from RewardBench to fine-tune Llama-3-8B on a subset of WildChat [Zhao et al., 2024] beats LLMs trained with the best RM and with a baseline that sequentially selects RMs, with $56.34\%$ and $71.45\%$ win rates (respectively) on length-controlled AlpacaEval [Dubois et al., 2024]. LASER also beats the ensemble of RM scores baseline with $72.69\%$ win rates using Llama-3-8B. Moreover, beyond the use in LLM training, our results demonstrate the effectiveness of LASER's RM selection strategy at inference time for long-form generation tasks in reranking LLM responses using multiple RMs; on LongBench [Bai et al., 2022], we find LASER beats the ensemble of RM scores baseline by 2.96 F1 points on single-document QA tasks and 2.97 F1 points on few-shot learning when using best-of-$n$ sampling for Llama-3-8B. Our analysis reveals that LASER is more efficient than sequential multi-RM and RM score ensemble baselines in terms of training time (wall-clock hours) by a factor of $3\times$, and $2\times$, respectively, while being more robust to conflicting preferences (Sec. 5).

## 2 Related Work

**Multiple Reward Ensembles.** Recent work explores training LLMs using multiple rewards, often via ensembles [Ramé et al., 2024, Wu et al., 2024, Coste et al., 2023, Zhang et al., 2024, Wang et al., 2024b, Jang et al., 2023, Eisenstein et al., 2023]. These methods typically aggregate or align scores across RMs, but suffer from inefficiency (both in terms of training and using multiple RMs) and conflicting rewards [Rita et al., 2024]. In contrast, LASER selects one pretrained RM per training step, avoiding these issues while outperforming ensemble-based baselines [Coste et al., 2023, Wang et al., 2024e]. Other approaches train an RM from interpretable objectives using a human-annotated dataset [Wang et al., 2024c] or jointly train multiple task-specific RMs with a sparse router [Quan, 2024]. Unlike these methods, LASER instead uses off-the-shelf RMs to train on LLM-generated outputs, which has shown better performance [Ivison et al., 2024], and dynamically selects RMs from a leaderboard using a multi-armed bandit without any RM training or annotated datasets.

**Iterative LLM Training.** Standard RLHF pipelines rely on static human preference data [Ouyang et al., 2022, Bai et al., 2022, Touvron et al., 2023], which limits scalability by the size and quality of annotated preference data and the effectiveness of off-policy optimization [Xu et al., 2023, Xiong et al., 2024, Yuan et al., 2024b, Guo et al., 2024]. Recent iterative methods improve this by generating feedback from model outputs, using gold labels [Singh et al., 2023, Pang et al., 2024], single RMs [Gulcehre et al., 2023], or self-judgment [Yuan et al., 2024b, Chen et al., 2024b]. LASER extends this by leveraging multiple RMs, avoiding pitfalls of single RM reliance (unreliable rankings, lack of generalization, and over-optimization), and reducing user burden in RM selection for a specific task [Huang et al., 2023].

**Multi-Armed Bandits (MABs).** MABs have been widely applied in ML for optimization and selection tasks [Chen et al., 2013, Li et al., 2010, 2018, Graves et al., 2017], including language models [Pasunuru et al., 2020, Krishnamurthy et al., 2024, Dwaracherla et al., 2024]. LASER uses MABs to dynamically select RMs during training, unlike prior work, which uses MABs to select the annotated samples for a single fixed RM [Dwaracherla et al., 2024] or uses MABs for model selection at test-time without RMs [Nguyen et al., 2024, Li, 2025]. This allows LASER to improve LLMs iteratively without gold labels or explicit optimization for evaluation metrics. To our knowledge, LASER is the first method to do that in the context of aligning LLMs with multiple RMs.

## 3 LASER: Learning to Adaptively Select Rewards

First, we expand on the single-iteration training pipeline with a general reward function (Sec. 3.1). Then, in Sec. 3.2, we describe how LASER dynamically selects an RM using MAB algorithms, assigning the RM for a given instance or batch. Finally, in Sec. 3.3, we describe cross-iteration training and how we update the MAB parameters. An illustration of LASER is shown in Fig. 1.

### 3.1 Training LLMs Using a Reward Function

LASER involves training with multiple RMs using a multi-armed bandit (MAB), which selects one model at a time. Therefore, we first describe how we train LLMs with generated data assuming a single RM; this corresponds to the top-right in Fig. 1 (in blue).

**Notation.** Following Yuan et al. [2024b] and Pang et al. [2024], we adopt an iterative training pipeline to fine-tune the LLM for $M$ iterations. Let $\pi_m$ be the LLM at iteration $m$; we assume that we start from an initial pretrained model $\pi_0$. Let $\mathcal{D} = \{x_1, x_2, \ldots, x_N\}$ represent the training inputs, where $x_i$ is an input query or prompt. Corresponding to each input query $x_i$, we sample a set of $n$ responses from the LLM at the current $m^{\text{th}}$ iteration as $\mathbf{y}_i = \{y_i^1, y_i^2, \ldots, y_i^n\} \sim \pi_m(y|x_i)$. Let $R^\star : (y_i^j | x_i) \to \mathbb{R}$ be a reward function that can score an LLM-generated response $y_i^j$ to a query $x_i$ based on how well it aligns with specific task objectives or instructions. Note that $R^\star(.)$ can be any reward function and may correspond to a single RM, one of the multiple RMs selected by the MAB (as in our case), or even the true reward.

**Generating Preference Pairs.** We evaluate each response $y_i^j$ using the reward function $R^\star(y_i^j | x_i)$. By comparing the rewards assigned to different responses, we can form $P$ preference pairs $(y_i^w, y_i^l)$,

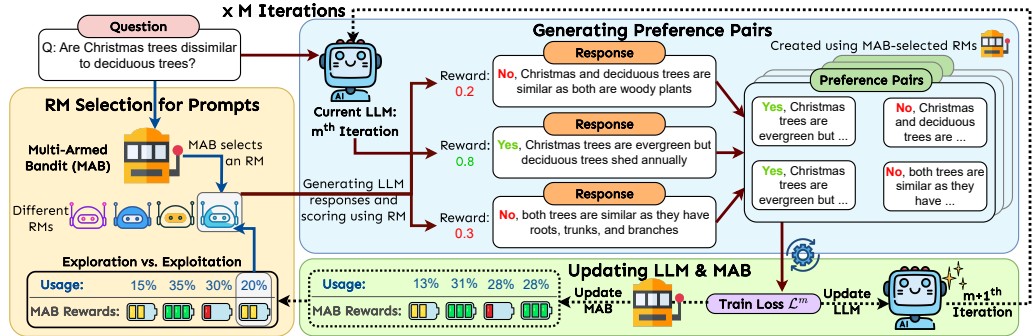

Figure 1: Overview of LASER. Given the query, the multi-armed bandit selects an RM depending on the underlying query and the bandit's parameters (based on the usage of each RM and the expected MAB reward). At iteration $m$, the LLM generates multiple responses that are scored based on the selected RM for that query. These responses are ranked into preference pairs, which are then used to fine-tune the model. The same training loss $\mathcal{L}^m$ is used to update the parameters of the LLM as well as the MAB for the next iteration, making the entire pipeline iterative.

where $y_i^w$ is preferred over $y_i^l$ if $R^\star(y_i^w|x_i) > R^\star(y_i^l|x_i)$, thereby building a preference dataset:[2]

$$\mathcal{D}_{\text{pref}} = \{(x_i, y_i^w, y_i^l) \mid x_i \in \mathcal{D}, R^\star(y_i^w) > R^\star(y_i^l)\}.$$

**Training Loss Function ($\mathcal{L}^m$).** In each iteration, we fine-tune the model using the generated preference dataset $\mathcal{D}_{\text{pref}}$, resulting in $M$ models $\pi_1, \pi_2, \ldots, \pi_M$. Specifically, we update the model using the DPO loss [Rafailov et al., 2024b] for learning from the preference pairs, which is consistent with other work on iterative LLM training [Yuan et al., 2024b, Pang et al., 2024]. In this work, we use the following loss functions:

$$\mathcal{L}_{\text{DPO}}^m(\pi_m) = -\mathbb{E}_{\mathcal{D}_{\text{pref}}} \left[ \log \sigma \left( \beta \left( r_i^w - r_i^l \right) \right) \right]; \mathcal{L}_{\text{NLL}}^m(\pi_m) = -\mathbb{E}_{\mathcal{D}_{\text{pref}}} \left[ \frac{\log \pi_m(y_i^w \mid x_i)}{|y_i^w|} \right],$$

where $r_i^w = \log \frac{\pi_m(y_i^w|x_i)}{\pi_{m-1}(y_i^w|x_i)}$, $r_i^l = \log \frac{\pi_m(y_i^l|x_i)}{\pi_{m-1}(y_i^l|x_i)}$, $\pi_m$ and $\pi_{m-1}$ denote the LLM in the current iteration $m$ and the previous iteration $m-1$ (used as the reference model in DPO loss). Following Yuan et al. [2024b], we use the standard DPO loss for instruction-tuning. Following Pang et al. [2024], we use the NLL loss on the preferred responses as an additional regularizer for reasoning tasks, i.e., $\mathcal{L}^m = \mathcal{L}_{\text{DPO}}^m + \mathcal{L}_{\text{NLL}}^m$. In Appendix C, we show that LASER outperforms baselines irrespective of the choice of the loss function $\mathcal{L}^m$.

## 3.2 Bandit Algorithms for Adaptive RM Selection

Sec. 3.1 described the data creation and LLM training procedure for our method when using a general RM (Fig. 1; top-right), which trains the LLM for a single mini-batch. Here, we describe the process by which we adaptively *select an RM for each batch of queries* using bandit algorithms (shown in Fig. 1-left, in yellow) and update the parameters of the bandit (more details in Appendix A.2).

**Background: Multi-Armed Bandits.** The multi-armed bandit (MAB) problem addresses the challenge of balancing exploration and exploitation in sequential decision-making [Vermorel and Mohri, 2005, Audibert et al., 2009]. The goal is to maximize cumulative MAB rewards over time by selecting arms that yield the highest MAB rewards. In order to distinguish between rewards or scores generated by RMs and the rewards used in MAB literature, we refer to the latter as "MAB rewards". A decision-making agent in a bandit setting faces a trade-off: whether to exploit the arm with the highest known MAB reward based on past observations or explore other, less familiar arms to gather more information that might lead to even better rewards in the future. In a contextual MAB setting,

---

[2]Following Pang et al. [2024], we randomly sample $P = 10$ pairs corresponding to each prompt $x_i$. For brevity, we omit this in the notation of $\mathcal{D}_{\text{pref}}$; but in our setting $|\mathcal{D}_{\text{pref}}| = P \times |\mathcal{D}|$.

the agent is also provided with additional information in the form of a context, such as the current state and input, to help inform arm selection accordingly.

**Challenges in Applying MAB to LLM Training.** Although MAB is a promising framework for selecting RMs in LLM training, several challenges remain. First, in typical MAB problems, the arms are often fixed with stable reward distributions. In contrast, during preference fine-tuning of LLMs, the arms are RMs whose outputs interact with and influence the training of the LLM itself. As the LLM is updated, the distribution of responses (and therefore the RM-derived reward signal) shifts. Second, unlike traditional MAB settings where rewards are observed directly, in our setting, no explicit supervision is available to indicate which RM is best per query. Finally, a static MAB setup struggles when new RMs are introduced or when existing RMs are noisy or domain-specific. We describe LASER in detail, explain how it incorporates MAB into LLM training, and addresses these challenges in the following section.

**Exploration and Exploitation of RMs.** LASER uses MABs to dynamically identify the most suitable RM for each query $x_i$ and task through exploration while simultaneously training the LLM (since this fine-grained information is not known a priori). Pulling a previously un(der)-explored arm allows the MAB to adaptively update its information about the relevance and quality of preference pairs built using that RM via the MAB reward (discussed below). If we over-explore RMs, we might waste time on underperforming RMs, slowing down the overall training progress. On the other hand, if we over-exploit, we might prematurely focus on one RM that seems best initially but is not optimal for all queries or tasks. The exploration-exploitation nature of MAB ensures that LASER adapts to the specific needs of each query (see Appendix B for further analysis) while improving the overall performance of the model.

**RM Selection in LASER.** LASER uses mini-batch training for each iteration, i.e., we use MABs to select a *single RM for a batch of prompts* $\mathbf{x}_{m,t}$ for $t^{\text{th}}$ batch or training step of iteration $m$ (total of $T$ steps/batches in each iteration).[3] Let the set of $K$ reward models (or arms) be denoted by $\mathcal{R} = \{R_1, R_2, \ldots, R_K\}$, where each $R_k$ corresponds to a different RM. We use the negative cumulative train loss function on the batch $(-\mathcal{L}^m)$ at the given iteration $m$, i.e., $\mathcal{L}^m = \mathcal{L}^m_{\text{DPO}} + \mathcal{L}^m_{\text{NLL}}$ for reasoning tasks, $\mathcal{L}^m = \mathcal{L}^m_{\text{DPO}}$ for other tasks, as mentioned in Sec. 3.1, as the MAB reward. The MAB reward is computed *after* the model is trained on each batch. Specifically, the MAB reward is the negative training loss (DPO), which depends on how clearly the model learns to prefer the RM's chosen outputs over the rejected ones. A lower DPO loss or a higher MAB reward corresponds to a larger log-likelihood margin between preferred and dispreferred responses, i.e., $\log \pi_m(y_i^w \mid x_i) - \log \pi_m(y_i^l \mid x_i)$ for a query $x_i$ and preference pair $(y_i^w, y_i^l)$, indicating that the RM's feedback helped the model sharpen its rankings and increase confidence in distinguishing between chosen and rejected responses. Thus, an RM that provides more informative and consistent rankings is given a higher reward compared to a less-suited RM providing uninformative rankings[4]. To further validate our MAB reward design, we empirically compare it with alternative learning signals for MAB in Appendix C. It is also worth noting that, since the MAB reward is computed based on preference pair data, LASER can treat any RMs, including learned RM signals or domain-specific metrics as part of its selection framework, thereby enabling generalization and adaptation to different types of RMs (see Appendix C for further analysis and experiments).

We employ LinUCB [Li et al., 2010], a contextual bandit algorithm for the arm or RM selection. We choose LinUCB because it is a contextual bandit algorithm that effectively leverages context information, making it well-suited for integration into LLM frameworks due to its ability to dynamically adapt decisions based on contextual embeddings. Additionally, LinUCB has been shown to provide a good trade-off between computational efficiency and performance [Zhou, 2015]. Therefore, it can be incorporated into any iterative LLM training framework with minimal overhead (described in detail in Appendix A.1). LinUCB assumes that the MAB reward can be modeled linearly as a function of context features and computes the expected MAB reward of each arm with an upper confidence bound to ensure exploration [Garivier and Moulines, 2008, 2011]. In each step $t$, we have a batch of input prompts $\mathbf{x}_{m,t}$ for which we compute sentence embeddings, using the policy model $\pi_m$, and use

---

[3]Note that LASER can switch between RMs at the instance level if the batch size is set to 1; however, for the sake of efficiency, we batch instances together both for LASER and the baselines, as this reduces the computational overhead associated with loading RMs onto the GPU. We provide further discussion in Appendix A.1.

[4]Note that since DPO training typically begins from a supervised fine-tuning (SFT) checkpoint, the policy model already has some ability to distinguish good responses from bad ones. In Fig. 5 we show empirically that the model's margin is typically positive, further supporting this claim.

the mean sentence embedding as the context $c(t)$ to the MAB, i.e., $c(t) = \sum_{x \in \mathbf{x}_{m,t}} e_m(x)/|\mathbf{x}_{m,t}|$, where $e_m(.) \in \mathbb{R}^d$ yields the sentence embedding from the model $\pi_m$. We calculate the embedding for a prompt as the last token embedding from model $\pi_m$. We use the last-token embedding as the context representation because in Transformer-based LLMs, this position typically aggregates information from the entire sequence. Moreover, this method is a standard and commonly used approach for extracting LLM embeddings [Wang et al., 2024d, BehnamGhader et al., 2024]. We provide more details regarding token embedding extraction in Appendix A.1 and an ablation study comparing context embedding methods in Appendix D. Even if individual batches vary, the repeated exposure to diverse inputs allows the bandit to learn which RMs are more helpful overall. Over training iterations, this appears to provide a robust signal for the bandit to learn effective RM selection, as demonstrated by LASER's consistent gains across datasets and the convergence behavior of MAB rewards shown in Appendix D.

The learned parameters of the LinUCB bandit include $\hat{\theta}_k \in \mathbb{R}^d$ which represents the learned weights for the features of each reward model and $A_k \in \mathbb{R}^{d \times d}$ (a covariance matrix) and a bias vector $b_k \in \mathbb{R}^d$ corresponding to each arm or RM $R_k$. We initialize the parameters for LinUCB by randomly initializing $b_k$ and setting parameter $A_k$ to the identity matrix. Based on the LinUCB algorithm, for each batch, the selected RM $R_t^\star$ is determined by $R_t^\star = R_j$ such that:

$$j = \arg \max_{k \in [1,K]} \left( c(t)^\top \hat{\theta}_k + \alpha \sqrt{c(t)^\top A_k^{-1} c(t)} \right), \tag{1}$$

where $\hat{\theta}_k = A_k^{-1} b_k$, and $\alpha$ is a parameter that controls the degree of exploration in LinUCB. A higher $\alpha$ encourages the algorithm to explore more aggressively by assigning higher uncertainty bonuses to actions with less information, while a lower $\alpha$ leads to more conservative behavior that prioritizes exploitation of known information. We provide an ablation study in Appendix D to examine the effect of $\alpha$ on the performance of LASER. $A_k$ and $b_k$ are updated based on the MAB reward for each RM, which corresponds to the normalized negative train loss $-\hat{\mathcal{L}}^m$ (described in detail in Appendix A.2):

$$A_k \leftarrow A_k + c(t)c(t)^\top; b_k \leftarrow b_k - \hat{\mathcal{L}}^m(t)c(t). \tag{2}$$

We emphasize that the effectiveness of our approach is *not* tied to particular bandit algorithm but rather from the *exploration-exploitation trade-off of MAB framework*. LASER is agnostic to the bandit algorithms and can incorporate alternatives to LinUCB. Indeed, we provide a comparison of LinUCB and other bandit algorithms in Appendix C.

### 3.3 LLM and Bandit Training in LASER

A key aspect of our approach is the generation of new preference training data in each iteration using the generations of the LLM itself and the RM selected by the MAB. Fig. 1 presents our training procedure, broken down into three stages: (i) the MAB selects an RM $R_t^\star$ (see Sec. 3.2; Fig. 1 left), generating preference pairs by scoring the LLM's outputs using the RM (Fig. 1 (top-right)), and parameter updates to the LLM and MAB. In this way, the model continuously learns from its own outputs, guided by the selected reward model. After each LLM train step (i.e., one mini-batch), the MAB's parameters are updated based on the observed MAB reward, i.e., how much the LLM's loss decreased from using the selected RM. In the case of LinUCB, this involves updating the parameter estimates $b_k, A_k$ (see Fig. 1; bottom in green). This entire process – selection of reward models, generation of new supervision data, fine-tuning, and bandit updates – repeats for a total of $M$ iterations (summarized in Algorithm 1).

**LASER with Best-of-$n$ Sampling.** For settings where fine-tuning the LLM is not desirable or feasible, LASER can also be applied to learn the MAB parameters without training the LLM. Rather than fine-tuning the model with preference data, we employ best-of-$n$ sampling [Lightman et al., 2023, Sun et al., 2024], where multiple responses are generated, and the best one is selected based on the RM. The bandit parameters are then updated using equation (2), with the MAB reward calculated as the negative normalized NLL loss on the train data. This updated bandit can subsequently be used for inference on the test set. This approach is particularly useful for long-context understanding tasks, where training would be too computationally intensive (see setting in Sec. 4.2).

# 4 Experiments and Results

In this section, we evaluate LASER across three domains: reasoning, instruction following, and long-context understanding. We compare our approach against baselines that utilize either a single RM or multiple RMs during LLM training. More detailed settings for both LASER and the baselines are provided in Appendix A.1, and additional experimental results are presented in Appendix B.

## 4.1 Experimental Setup

**Models.** We conduct our experiments on the Llama-3-8B base [AI@Meta, 2024] model. We present additional results with Mistral-7b-v3-Instruct [Jiang et al., 2023] and Qwen2.5-32B [Qwen Team, 2024a] in Appendix B. For training, all models are fine-tuned using Low-Rank Adaptation (LoRA) [Hu et al., 2021] for efficiency. For both training and inference, we do 0-shot prompting and sample $n = 30$ responses per prompt with temperature $0.8$ (see Appendix A.1 for more details).

**Reward Models.** We select $K = 4$ strong 7B RMs from RewardBench [Lambert et al., 2024], which include Zephyr-7B-Alpha, Qwen1.5-7B-Chat, Eurus-7B-KTO, and OLMo-7B-Instruct. Following the pipeline outlined in Lambert et al. [2024], for these models, we compute the reward for each response as the log-likelihood of the RM for that response (see Appendices A.1 and E for more details and discussion about the RM choices).

**Datasets and Metrics.** Our experiments cover a range of tasks and datasets (see Appendix A.1):

- **Reasoning:** Evaluating reasoning abilities is crucial for testing the model's capacity to handle complex, multi-step tasks and has presented a challenge to iterative preference optimization methods [Yuan et al., 2024b, Chen et al., 2024b]. We train and evaluate on StrategyQA [Geva et al., 2021], MMLU [Hendrycks et al., 2021b,a], and GSM8K [Cobbe et al., 2021].
- **Instruction-Following:** We further evaluate our method on heterogeneous tasks without gold labels. We use user prompts from WildChat dataset [Zhao et al., 2024], which contains a collection of natural user-chatbot interactions. This dataset has five primary categories of instruction-following prompts: creative writing, analysis, coding, factual information, and math reasoning. Due to computational constraints, we randomly subsample 5K prompts from each category for model training. We compare models trained with LASER against baselines (described below) using length-controlled AlpacaEval [Dubois et al., 2024] that pairs responses from two different LLMs and uses GPT-4o as a judge to pick the winner, accounting for the length of both responses.
- **Long-Context Understanding:** As fine-tuning LLMs on long-context inputs is computationally intensive, we demonstrate the effectiveness of LASER using Best-of-$n$ sampling on LongBench [Bai et al., 2023], which consists of multiple tasks, such as single-document QA, multi-document QA, summarization, and few-shot learning. For summarization, we report Rouge-L [Lin, 2004], whereas for the rest, we report the F1 score.

**Baselines.** We compare LASER against two baseline categories reflecting different kinds of methods for using RMs in LLM training: single RM selection and multi-RM ensemble baselines. For single RM selection baselines, we evaluate against the following baselines:

- **Best RM**: From our collection of RMs, we pick the RM that corresponds to the best overall score on RewardBench [Lambert et al., 2024]: Zephyr-7B-Alpha. We use this *single* RM during training (c.f. Sec. 3.1). This baseline reflects the performance gain a user could expect when selecting the best RM from a leaderboard without knowing *a priori* how it generalizes to a particular task.
- **Avg. RM**: Here, we perform single RM training over all the RMs in the pool and report the average performance. A comparison with this baseline represents an expected gain from a randomly picked RM from a leaderboard; however, note this baseline consists of multiple models averaged together.
- **Random RM Selection:** In this baseline, we randomly sample a single RM from the set of RMs (from a uniform distribution) for each training batch in every iteration.
- **Sequential RM Selection**: In training, this method explores RMs sequentially and based on a set order in each iteration to examine their impact on model training, demonstrating that, instead of optimizing with all RMs, LASER can adaptively select the best RM for each batch of samples.
- **Classifier Selection:** To compare against a context-sensitive baseline that does not use an MAB, we train a $K$-way classifier to perform RM selection using data from RewardBench (see Appendix A.1). Specifically, for each query and RM, we compute the RM's score of the annotated preferred and dispreferred response. The RM that assigns the correct preference ordering with the highest

difference between the scores of the preferred and dispreferred responses is chosen as the *RM label* and used to train the classifier. While training the LLM, for each input $x_i \in \mathcal{D}$, we select the RM for building preference pairs based on this trained classifier.

Additionally, we compare against the following RM ensemble baselines:

- **RM Score Ensemble:** We generate multiple responses for each query, which are then scored using each RM, and the preference dataset is created by averaging the scores across all RMs (following Coste et al. [2023]); thus, comparing LASER with using all RMs simultaneously.

- **RM Agreement Ensemble:** Because ensembling scores through averaging is sensitive to the absolute scores produced (which may differ between RMs), we follow Wang et al. [2024e] in ensembling through ranking and agreement. Specifically, we generate 32 responses for each query and sample 100 pairs for each. We score each pair with RMs, constructing a preference dataset by choosing the 10 pairs with the highest agreement of preference rankings across RMs.

- **Online RM Ensemble:** Since uniformly-ensembled RM scores might fail to prevent the LLM from exploiting an underperforming RM, we further compare our method to an additional ensemble baseline that uses an online learning approach. Instead of setting uniform weights for each RM as in RM Score Ensemble, each RM is assigned a learned weight; these weights are dynamically updated after each training batch using the multiplicative weights algorithm [Arora et al., 2012].

Conceptually, among the single RM selection methods, the best RM baseline serves as an "exploit-only" setting that exploits the best available RM based on aggregate RewardBench scores. On the other hand, the random and sequential selection baselines are "explore-only" in that they pick a new RM either randomly or via a predefined sequence, irrespective of the performance of each arm (RM). LASER and the classifier approach represent two methods for selecting a single optimal RM for each instance; however, we note that the classifier approach depends on a fine-grained, in-distribution dataset of queries paired with their corresponding suitable RMs for annotating preferences, which is often absent in practice. In our experiments, we demonstrate the benefits of balancing the exploration-exploitation trade-off compared to "exploit-only" and "explore-only" approaches. Additionally, different RM ensemble baselines employ various methods for combining RMs while utilizing them simultaneously. Furthermore, we analyze potential conflicting signals among RMs to highlight the advantages of our framework over RM ensemble baselines (Sec. 5).

## 4.2 Main Results

**LASER achieves the best average accuracy on reasoning tasks.** Table 1 demonstrates that our method consistently outperforms the baselines across multiple reasoning benchmarks, particularly in the StrategyQA and GSM8K datasets. For example, LASER improves by approximately 2% absolute accuracy over the sequential baseline on both GSM8K and StrategyQA datasets.

In cases where the best RM is not known beforehand, LASER surpasses the performance of the average RM baseline by 2.7% and the RM Score Ensemble baseline for each instance by 2.67% (in accuracy averaged over the three datasets). Moreover, this lower performance by the RM Score baseline is not purely due to variance in the scores: LASER also surpasses the RM Agreement Ensemble and RM Online Ensemble by 0.91% and 2.27%, respectively. Overall, LASER provides *consistent* results while the underlined second-place models show *inconsistent* performance across datasets. These results also emphasize the advantages of LASER, as it eliminates the need to choose a different RM in advance or ensemble multiple RMs.

Table 1: Performance on reasoning benchmarks on Llama-3-8B. The baselines also include supervised fine-tuning on human-written responses (SFT) as a reference for performance without preference optimization. The highest accuracy is shown in bold, and the second-highest accuracy is underlined. LASER yields the highest average accuracy.

| Method | StrategyQA | GSM8K | MMLU | Avg. |
|---|---|---|---|---|
| SFT | 80.41 | 69.43 | 65.66 | 71.83 |
| Best RM | 84.29 | 73.16 | 67.15 | 74.87 |
| Avg. RM | 82.62 | 71.57 | 66.67 | 73.62 |
| Random RM Selection | 84.37 | 71.99 | 67.85 | 74.74 |
| Seq. RM Selection | 83.90 | 72.94 | 68.02 | 74.95 |
| Classifier Selection | 83.13 | 72.73 | 67.96 | 74.60 |
| RM Score Ensemble | 82.96 | 70.94 | 67.04 | 73.65 |
| RM Agree. Ensemble | 84.03 | 73.85 | 68.35 | 75.41 |
| RM Online Ensemble | 83.25 | 72.04 | 66.85 | 74.05 |
| LASER (Ours) | 85.96 | 74.75 | 68.24 | 76.32 |

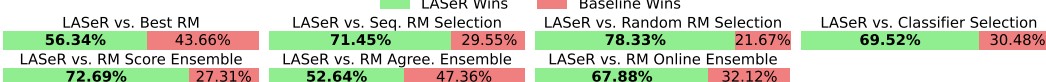

Figure 2: Length-controlled AlpacaEval win rates comparing LASER against baselines on WildChat instruction-following tasks using Llama-3-8B. The top row shows comparisons against single RM selection methods, while the bottom row shows comparisons against multi-RM ensemble methods.

**LASER beats baselines at instruction-following.** Often, LLMs are used by large numbers of people with a diverse set of queries, goals, and intentions, and their preferences vary based on the underlying query. To demonstrate the effectiveness of LASER in such settings, we compare the instruction-following performance in Fig. 2, i.e., AlpacaEval win rates, of LLMs trained using LASER with the baselines using WildChat. Specifically, LASER achieves substantial win rates compared to single RM selection baselines such as random and sequential selection, with 78.33%, and 71.45%, respectively. We also outperform training with the single best RM (per RewardBench) by a 56.34% win rate. Compared to RM ensemble baselines, LASER achieves 72.69% and 67.88% win rates over RM Score Ensemble and RM Online Ensemble, respectively. We hypothesize the lower win rate of the baselines stems from the inability of these baselines to deal with conflicting signals from multiple RMs (see Fig. 3 for further analysis). Overall, these results highlight that LASER excels without gold labels and performs consistently well at following instructions across various user queries, showcasing its adaptability to diverse tasks.

**LASER's adaptive RM selection helps long-context understanding.** Given the cost of training long-context systems, for LongBench [Bai et al., 2023], rather than fine-tuning a model using RMs, we employ the selected RM to rerank generation in Best-of-$n$ sampling (see Sec. 3.3).

In Table 2, we observe that LASER consistently outperforms the baselines across tasks. LASER improves single-doc QA (SiQA) by 3.58 F1 points over the base Llama-3-8B model and 2.64 F1 points over random RM selection. On multi-doc QA (MuQA), our approach improves performance over the Llama-3-8B base model by $\approx 4$ F1 points. Furthermore, on few-shot (FS) learning tasks, LASER provides over 3 points gain in F1 compared to the base model, surpassing the average RM performance by up to 2.4 F1 points and demonstrating its effectiveness across tasks. Lastly, Table 2 demonstrates that LASER consistently outperforms the RM Score Ensemble baseline across different long-context tasks (except for the Summarization task where LASER is comparable to Best RM), e.g., a $\approx 3$ F1 point boost on single-doc QA and few-shot learning tasks.

Table 2: LASER outperforms baselines in long-context understanding tasks with Llama-3-8B. Sequential RM selection is not applicable in this setting, as only inference is conducted. For QA and few-shot learning tasks, we report F1 scores, and for summarization, we report Rouge-L.

| Method | SiQA | MuQA | Sum | FS |
|---|---|---|---|---|
| Base model | 33.89 | 32.96 | 29.54 | 70.23 |
| Best RM | 35.12 | 35.83 | **34.26** | 71.79 |
| Random RM Selection | 34.83 | 35.19 | 31.57 | 70.91 |
| Classifier RM Selection | 34.42 | 34.24 | 32.41 | 70.58 |
| RM Score Ensemble | 34.51 | 35.52 | 32.38 | 70.34 |
| RM Agree. Ensemble | 35.79 | 35.41 | 33.19 | 72.15 |
| RM Online Ensemble | 34.69 | 35.80 | 32.89 | 70.51 |
| LASER (Ours) | **37.47** | **36.94** | 34.13 | **73.31** |

## 5 Additional Analysis of LASER

In this section, we present analyses on the presence of conflicts among RMs, which helps justify the benefits of LASER, as well as its training efficiency compared to the baselines. We further provide additional analyses, including the generalization ability of LASER in Appendix C, and ablation studies on LASER's design in Appendix D.

**Presence of Conflicting Signals among RMs.** In Sec. 4.2, we find that LASER consistently outperforms other RM ensemble baselines across a wide variety of tasks. We attribute some of these performance gains to the inability of the multi-RM baseline to handle conflicting signals, resulting in subpar training data from multiple RMs. To study this, we sample pairs of outputs generated by Llama-3-8B on MMLU as well as WildChat and evaluate the consistency of response preferences measured by multiple RMs. Since pair-wise preferences are binary, we compute F1 to measure consistency with one

RM's preferences serving as the reference. Fig. 3 (left on MMLU) reveals that Qwen and Zephyr have the highest agreement rate at $0.77$, while Qwen's agreement with Eurus and Olmo is lower at $0.58$ and $0.43$, respectively. This is expected as Qwen and Zephyr are the top-performing models in reasoning according to RewardBench, while Olmo ranks the lowest in reasoning ability among the four models.

We observe similar trends in agreement across RMs on WildChat (albeit with different agreement scores), which contains user queries asked LLMs in the wild; see Fig. 3 (right). It appears that for more heterogeneous datasets with more categories, the level of disagreement among RMs (or conflict) increases. This also highlights LASER's advantages over multi-RM baselines that do not address conflicts in RMs and may explain why choosing one RM in LASER and the best RM baseline outperforms multi-RM ensembles.

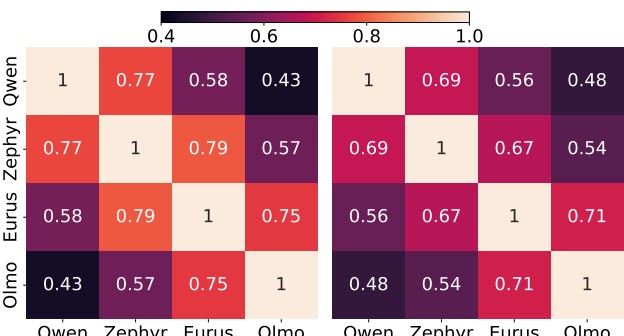

Figure 3: Agreement in preference rankings between RMs on MMLU (left) and WildChat (right).

**Training Efficiency of LASER.** We show the accuracy-training time tradeoff in Fig. 4 by comparing the GSM8K performance of training with LASER and different baselines, along with the corresponding wall clock training time. Wall clock time is measured as the training time of a model (hours), keeping compute resource consistent. We find that sequentially optimizing over each RM performs the worst in terms of training time ($3\times$ of LASER) while RM score ensemble has the worst accuracy (and takes $2\times$ the training time of LASER). Moreover, LASER outperforms all other baselines in terms of accuracy while maintaining the lowest training time, being more than twice as fast as the second-best baseline. LASER's efficiency comes from fast convergence compared to the sequential selection baseline and avoiding the overhead of loading and evaluating multiple RMs per step, as required in RM ensemble methods (Online Ensemble, Score Ensemble, and RM Agreement Ensemble). Instead, it selects and uses only one RM per mini-batch, saving GPU memory and compute. Ensemble baselines require scoring all candidate responses with all RMs, whereas LASER scores responses with only the selected RM, reducing overhead. Lastly, we note that LASER can be run in an offline data creation mode, where RM selection and scoring are done once to generate preference data, unlike online ensemble methods that require multiple RM passes per step.

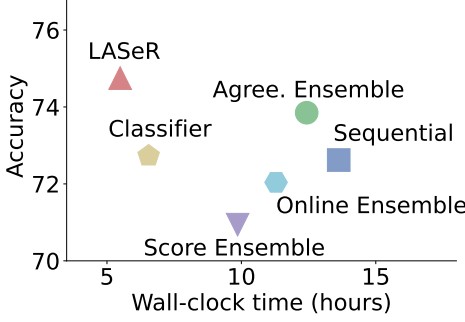

Figure 4: Training efficiency of LASER vs. different baselines on GSM8K.

## 6 Conclusion

We present LASER, an adaptive method for selecting RMs and iteratively training LLMs using multiple RMs. We formulate the problem as a contextual multi-armed bandit problem, learning to select the RM that most improves the LLM conditioned on the given input or query. We test LASER across diverse settings, showing its utility on reasoning tasks, instruction-following tasks, and long-context generation. Across domains, we show that LASER *consistently* results in superior performance, whereas multi-RM baselines that select RMs using random or fixed strategies or ensemble multiple RMs uniformly have lower and more variable performance. In our analysis, we show that LASER is robust to noisy RMs, and flexibly uses different RMs depending on the domain, and generalizes to multiple settings. Lastly, by selecting one RM at a time, LASER provides the best of both worlds: consistently outperforming all baselines while still maintaining efficiency by only optimizing for one model at a time.

## Acknowledgements

We thank the reviewers and Justin Chih-Yao Chen for their useful feedback. This work was supported by DARPA ECOLE Program No. HR00112390060, NSF-AI Engage Institute DRL2112635, DARPA Machine Commonsense (MCS) Grant N66001-19-2-4031, the Accelerating Foundation Models Research Program, NSF CAREER Award 1846185, Apple PhD Fellowship, and Capital One Research Award. The views contained in this article are those of the authors and not of the funding agency.

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

## Limitations and Broader Impacts

**Limitations.** While LASER demonstrates strong empirical performance and generalization across tasks, policy models, and RMs, it still has some limitations. First, like all work using RMs to improve LLMs, it relies on the availability of a high-quality pool of RMs, and downstream performance may degrade if the pool contains only weak or biased models. Additionally, although the bandit-based approach mitigates reliance on a single RM, it does not explicitly address issues of safety, fairness, or bias in RM selection and consider potential trade-offs between safety and task effectiveness when choosing an RM. Finally, besides the MAB formulation, other methods such as reinforcement learning-based RM selection might be applicable. This could allow for more flexible RM selection policies, but would introduce substantial optimization and stability challenges, especially when dealing with non-stationary and noisy RM feedback. We consider these directions as promising for future research.

**Broader Impact.** LLMs have been shown to reflect stereotypes, biases, and other negative traits contained in their pretraining data [Weidinger et al., 2021]. Consequently, fine-tuned LLMs (including those trained with LASER) may also exhibit such undesirable traits in their generations during inference or training and exhibit the same potential for misuse as any other fine-tuned model. While prior work has made some headway in detecting such harmful content generated by LLMs [Inan et al., 2023], considerable research effort is needed in mitigating bias in LLMs. Conceptually, classifiers that detect risky, harmful, or biased content in the text can also be used as an additional RM in LASER's training to reinforce avoiding bias via preference optimization. However, we do not study this in our work and leave it to future work to explore these directions.

## A Experiments

### A.1 Experimental Setting

**Training Setup.** We provide the training details for our method and baselines as follows:

- **LoRA:** For training with LoRA, we set the rank to 16 and alpha to 32.
- **Preference pairs construction:** Following Pang et al. [2024], we generate $P = 10$ pairs per query for training with our loss in Sec. 3.1.
- **Training iterations:** For all experiments, we trained each method to converge. The number of iterations is selected based on the observed convergence, with a performance metric threshold of 0.1 across training batches on the dev set. In particular, LASER, "Best RM", "Avg. RM", "Classifier RM", and RM ensemble baselines were trained for 10 iterations. For both the sequential and random RM selection, we found LLM training took longer to converge, and consequently, the model was trained for 25 iterations.
- **Batch size:** We fine-tune the model using a learning rate of $5e-6$ and a batch size of 16. Noted that we experimented with different batch sizes to evaluate their impact. Using a batch size of 1 yielded comparable performance to a batch size of 16 but was significantly less efficient in training the LLM due to the increased computational overhead. Based on this, we opted to use a batch size of 16 for a better trade-off between performance and efficiency. For datasets like Wildchat, which contain clearly defined and diverse categories, we structure the batches such that each batch consists of data belonging to a single category. This setup minimizes the risk of mismatches between the RM and the batch data, as each RM is evaluated on its most relevant data category. For reasoning datasets where such predefined categories do not exist, shuffling the data during training ensures diverse data and a good training signal for LLM within each batch. Even if individual batches vary, the repeated exposure to diverse inputs allows the bandit to learn which RMs are more helpful overall. Over training iterations, this appears to provide a robust signal for the bandit to learn effective RM selection, as demonstrated by LASER's consistent gains across datasets and the convergence behavior of MAB rewards shown in Appendix B.
- **Resources:** The LinUCB algorithm has a total of 1.6M learnable parameters (including matrix $A$ and bias vector $b$). Regarding the computation of MAB parameters, empirically, we find that for query embedding with dimension 4096 and using 4 reward models, inverse computation needs to be performed for each RM once per batch (and can be cached to be reused later), which only adds

a latency of 1.27 seconds. In comparison, a forward and backward pass of the LLM takes 36.39 seconds. Our experiments are run on 4 RTX A6000 with 48G memory each.

**RewardBench.** Following Lambert et al. [2024], rewards are computed with no reference model and only use the log-likelihood of the reward model. For instance, given a reward model $\pi_{R^\star}$, the reward for an input $x_i$ and response $y_i$ is calculated as: $\log \pi_{R^\star}(y_i \mid x_i)$. There is no need for normalization since we use this log-likelihood to rank the responses.

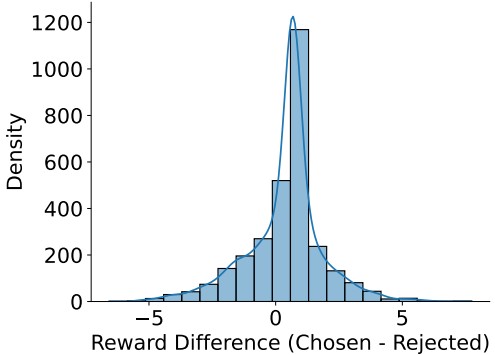

Figure 5: Reward difference distribution of chosen and rejected responses of Llama-3-8B on Reward-Bench.

To test the hypothesis that the log probabilities of the chosen and rejected responses from the policy model are generally close in value, with the chosen response tending to have a slightly higher log probability, we conduct an experiment using Llama-3-8B and RewardBench dataset. For each instance, we calculated the reward difference by subtracting the log probability (reward) of the chosen response from that of the rejected response. Formally, given a policy model $\pi_0$, a prompt $x_i$ and a pair of chosen response $y_i^w$ and rejected response $y_i^l$ from RewardBench data, we calculate the reward difference as $\log \pi_0(y_i^w \mid x_i) - \log \pi_0(y_i^l \mid x_i)$. As shown in Fig. 5, the distribution is centered near zero, indicating that the values are often close. However, the mass is skewed slightly to the right, suggesting that the chosen response tends to have a higher log probability than the rejected one in the majority of cases. This suggests the policy model generally has a reasonable ranking, but benefits from DPO's maximization of the margin.

**Details of RMs.** We provide details for each chosen RM:

- Zephyr-7B-Alpha: is a fine-tuned version of Mistral-7B model that was trained on on Ultra-Chat [Ding et al., 2023] and UltraFeedback [Cui et al., 2023] using DPO.
- Qwen1.5-7B-Chat: is pretrained with human-style conversation data inspired by Ouyang et al. [2022] along with questions, instructions, and answers in natural language, and post-trained with both SFT and DPO using diverse prompts [Lu et al., 2023].
- Eurus-7B-KTO: is a fine-tuned version of Eurus-7B-SFT model using KTO loss on UltraInteract [Yuan et al., 2024a] and UltraFeedback [Cui et al., 2023].
- OLMo-7B-Instruct: is the instruct version of OLMo-7B base model and was fine-tuned using UltraFeedback [Cui et al., 2023].

**Extracting Embeddings for a Query Using $\pi_m$.** To extract embeddings for a query using $\pi_m$, we first process the input query through the policy model $\pi_m$. We use the embedding of the last token in the query as the representation for the query. The embedding is then used as input to the subsequent bandit algorithm.

**Datasets.** For StrategyQA, GSM8K, and MMLU, we divided each dataset into training and test sets. The model is fine-tuned on the training set and dev set and then evaluated on the test set. For WildChat, the dataset was split into a 70/10/20 ratio for training, development, and testing. Following Zhao et al. [2024], prompt categorization is done using a small off-the-shelf classifier.[5] For LongBench, we subsample 5K examples for three tasks: multi-document QA, summarization, and few-shot learning.

---

[5]Link: `https://huggingface.co/valpy/prompt-classification`

Table 3: Number of examples in train, val, and test sets.

| Task | Dataset/Category | Train | Dev | Test | Total |
|------|-----------------|-------|-----|------|-------|
| Reasoning | StrategyQA | 1946 | 278 | 556 | 2780 |
| | GSM8K | 6750 | 750 | 1000 | 8500 |
| | MMLU | 11135 | 1591 | 3182 | 15908 |
| WildChat | Creative | 3500 | 500 | 1000 | 5000 |
| | Analysis | 3500 | 500 | 1000 | 5000 |
| | Coding | 3500 | 500 | 1000 | 5000 |
| | Factual | 3500 | 500 | 1000 | 5000 |
| | Math | 3500 | 500 | 1000 | 5000 |
| LongBench | Single-doc QA | 3534 | 505 | 1010 | 5049 |
| | Multi-doc QA | 3500 | 500 | 1000 | 5000 |
| | Summarization | 3500 | 500 | 1000 | 5000 |
| | Few-shot learning | 3500 | 500 | 1000 | 5000 |

Each category was split into a 70/10/20 ratio, and the bandit model was trained and validated on the training and development sets and then tested on the test set. We report the detailed number of instances for train, development, and test sets in Appendix A.1.

**Baselines.** Here we provide more details for baselines:

- **Classifier Selection.** We add an additional baseline that uses the RewardBench data to train a classifier that maps queries to an RM $C : \mathbb{R}^d \to \mathcal{R}$, where $\mathcal{R} = R_1, R_2, \ldots, R_K$ is the set of RMs. Specifically, to construct a dataset for training $C$, we take each query in the RewardBench data along with its corresponding chosen and rejected responses. The RewardBench dataset contains a total of 2985 examples across several categories, including chat, safety, and reasoning. The dataset is split into an 80/20 ratio for training/development sets, then the classifier is trained on the training set and validated on the development set. We use each RM to score these responses. The RM that assigns the correct score with the highest difference between the chosen and rejected response is selected to label the RM for that query. After training $C$, we use this classifier to select the RM used for training the LLM in our pipeline. In the experiments, we use a three-layer MLP with hidden dimensions of 2048 and 1024 and an output dimension of 4 (number of RMs), with ReLU activation in each layer.
- **RM Ensembles.** While the ensemble methods generate scores from multiple RMs in a single iteration for a fixed set of responses sampled at the start of the iteration, we still generate new responses at each training iteration as part of the overall learning process. This ensures that the training dynamically incorporates updated responses from the LLM.

**Licenses for Models and Datasets.** Below is the license for models and datasets used in this paper:

- LLMs: Llama-3-8B (Llama 3 Community License), Mistral-7B (Apache 2.0 License), Qwen2.5-32B (Apache 2.0 License).

- Datasets: StrategyQA, GSM8K, MMLU (MIT License), WildChat (ODC-BY License), LongBench (Apache 2.0 License).

**Additional Results and Analysis.** In Appendix B, we present additional analyses on the statistical stability of LASER and baselines across multiple runs (Table 4). We show that the RM selected by LASER adapts to input queries (Fig. 6) and aligns well with RewardBench ground truth rankings (Fig. 8). We also show that MAB rewards converge over training (Fig. 9), and that the learned MAB policy can be reused to improve cold-start training (Table 6). Detailed comparisons against individual RMs used in isolation (Tables 8 and 9) further confirm that LASER consistently outperforms all single-RM baselines. In addition, we show that LASER maintains strong performance across different sets of RMs (Table 10).

In Appendix C, we evaluate the generalization of LASER across multiple dimensions: different bandit algorithms (Table 11), base models and RMs (Tables 12 and 14), number of RMs (Fig. 11), training loss functions (Table 17), and out-of-distribution tasks (Table 18). We also show that LASER can successfully select evaluation metrics tailored to domain-specific reasoning tasks [Golovneva

et al., 2022, Prasad et al., 2023] (Table 19). Finally, LASER demonstrates robustness to both noisy RMs and underperforming reward signals (Figs. 12 and 13).

## A.2 Details of Bandit Algorithms

**Algorithm for Sec. 3.3.** We provide the detailed algorithm for Sec. 3.3 in Algorithm 1.

---

**Algorithm 1** Bandit-based Reward Model Selection for LLM Training

---

1: **Input:** LLM $\mathcal{M}$, reward models $\mathcal{R} = \{R_1, R_2, \ldots, R_K\}$, dataset $\mathcal{D} = \{x_1, x_2, \ldots, x_N\}$, bandit algorithm (LinUCB)
2: **Initialize:** Bandit algorithm parameters (e.g., $\theta_k$ for each RM)
3: **for** each training iteration $m = 1, 2, \ldots, M$ **do**
4:     **for** each batch or train step $t = 1, 2, \ldots, T$ **do**
5:         Select reward model $R_t^\star$ for time step $t$ using equation (1) (LinUCB)
6:         Sample a batch of samples from $\mathcal{D}$ and generate preference pairs following 3.1
7:         Fine-tune $\pi_m$ using preference pairs in $\mathcal{D}_{\text{pref}}$ using $\mathcal{L}^m$
8:         Update bandit parameters based on equation (2) (LinUCB)
9:     **end for**
10: **end for**

---

**Exp3.** Exp3 is a non-contextual bandit algorithm designed for adversarial settings. It maintains a probability distribution over the arms and selects arms based on the exponential weighting of past rewards. The probability for choosing arm $a_k$ at round $t$ is calculated as follows:

$$p_k(t) = (1 - \gamma)\frac{\exp(S_k(t))}{\sum_{a_k \in \mathcal{A}} \exp(S_k(t))} + \frac{\gamma}{K},$$

where $S_k(t)$ is the cumulative score for arm $a$ up to time $t$ and $\gamma$ is a parameter controlling the exploration rate.

The arm $a_k$ is selected by sampling the following categorical distribution

$$a_t \sim \text{Categorical}(p_1(t), \ldots, p_K(t)) \tag{3}$$

The score for arm $a_t$ is updated based on the observed normalized reward $-\hat{\mathcal{L}}^m(t)$ and the probability $p_k(t)$ of selecting that arm:

$$S_k(t + 1) = S_k(t) - \frac{\hat{\mathcal{L}}^m(t)}{p_k(t)} \cdot \mathbb{K}(a_t = a_k), \tag{4}$$

where $\mathbb{K}(a_t = a_k)$ is an indicator function that equals 1 if arm $a_k$ was selected at time $t$, and 0 otherwise.

**MAB reward normalization.** To maintain a consistent scale and magnitude of MAB rewards across training, we apply scaled rewards based on the quantiles of the reward history, following the method outlined by Graves et al. [2017]. Let $L = \{-\mathcal{L}^m(1), \ldots, -\mathcal{L}^m(t - 1)\}$ represent the unscaled reward history up to time step $t$. This history's lower and upper quantiles are denoted as $q_t^{lo}$ and $q_t^{hi}$, respectively. We set $q_t^{lo}$ and $q_t^{hi}$ to be $20^{th}$ and $80^{th}$ quantiles. The scaled reward, $-\hat{\mathcal{L}}^m(t)$, becomes:

$$-\hat{\mathcal{L}}^m(t) = \begin{cases} 0 & \text{if } -\mathcal{L}^m(t) < q_t^{lo} \\ 1 & \text{if } -\mathcal{L}^m(t) > q_t^{hi} \\ \frac{-\mathcal{L}^m(t) - q_t^{lo}}{q_t^{hi} - q_t^{lo}} & \text{otherwise.} \end{cases}$$

We chose to use the cumulative MAB reward because it provides a more comprehensive measure of how well the algorithm performs over time. In our framework, cumulative MAB reward reflects the performance of the model across batches and iterations, capturing the long-term impact of both exploration and exploitation decisions. Algorithms like LinUCB are designed to optimize cumulative rewards, as they ensure a balanced trade-off between exploration (gathering information about less-tested RMs) and exploitation (using the best-known RM).

# B Additional Empirical Results

**Analysis on Statistical Significance of LASER.** While our main results are based on single runs, we note that this is standard practice in iterative preference optimization setups, including recent works such as Pang et al. [2024], where large-scale LLM training limits the feasibility of multi-seed replication across all benchmarks. Nonetheless, to provide some insights regarding the statistical significance of LASER, we report the mean and standard deviation over 5 seeds on reasoning tasks. We compare LASER against the Best RM and RM Agreement Ensemble (identical random seeds to ensure fair comparison), which are the most competitive across our benchmarks. Table 4 shows that LASER consistently outperforms the baselines across runs, with standard deviations below 0.3% (lower than RM Agreement Ensemble), showing stable performance. We believe these consistent gains across tasks and models support the claim that LASER is not only superior but also reliable in expectation. To further support these results, we conduct a non-parametric hypothesis test. Specifically, we evaluate the following hypotheses: Null hypothesis: LASER's accuracy is equal to that of the competing method; Alternative hypothesis: LASER's accuracy is greater than that of the competing method. We use a one-sided non-parametric Wilcoxon signed-rank test on paired accuracy values across random seeds. The p-values of the tests between LASER and the two strongest baselines are reported in Table 5. At a significance level of 0.05, the results in Table 5 indicate that we can reject the null hypothesis, demonstrating that LASER significantly outperforms both baselines across all three tasks.

Table 4: Mean and standard deviation across methods on reasoning tasks.

| Method | StrategyQA | GSM8K | MMLU |
|---|---|---|---|
| Best RM | $\underline{84.26} \pm 0.09$ | $73.10 \pm 0.22$ | $67.11 \pm 0.13$ |
| RM Agree. Ensemble | $84.05 \pm 0.26$ | $\underline{73.65} \pm 0.42$ | $\underline{68.07} \pm 0.44$ |
| LASER | $\mathbf{85.87} \pm 0.11$ | $\mathbf{74.95} \pm 0.27$ | $\mathbf{68.32} \pm 0.26$ |

Table 5: Statistical significance (p-values) of LASER and baselines on reasoning tasks.

| LASER vs. Baseline | StrategyQA | GSM8K | MMLU |
|---|---|---|---|
| LASER vs. Best RM | 0.036 | 7e-4 | 0.019 |
| LASER vs. RM Agree. Ensemble | 8e-3 | 0.024 | 0.046 |

**LASER's Selected RM Adjusts to the Query.** Fig. 6 shows the relative utilization rates of each arm (i.e., RM) of the bandit on WildChat. We observed vastly different RM utilization rates depending on the underlying query within the *same dataset*. On queries requiring creativity in LLM responses, we find that Olmo and Eurus RMs are utilized about 20% more often than Qwen RM, despite Qwen RM being ranked higher on RewardBench. This can be explained by the fact that the Qwen RM largely underperforms on the "chat" subsplit of RewardBench (behind Olmo and Eurus by nearly 40 points in chat score). On the other hand, Qwen RM is used roughly half the time for user prompts involving math, while Olmo and Eurus are used sparingly. This is consistent with Qwen RM's ranking on the "reasoning" split of RewardBench, outperforming Eurus and Olmo RMs by 15-20 points. Note that LASER *automatically* deduces these relative rankings of RMs and uses them depending on the underlying query *without having access to the RewardBench leaderboard*. Therefore, RM utilization of LASER can serve as an analysis tool for future work when assessing performance on untested domains.

We observe a similar trend in LASER's RM selection on LongBench. We observe distinct utilization patterns for the QA tasks vs. summarization and few-shot learning. QA tasks exhibit nearly equal utilization of the top-2 RMs on RewardBench (Zephyr-7B-Alpha and Qwen1.5-7B-Chat in decreasing order), with the utilization of the Qwen RM even *exceeds* that of Zephyr RM for multi-document QA. In contrast, on summarization and few-shot learning the top RM (Zephyr) is far more preferred by LASER with margins of 59% and 31% over the second-best RM and the least performant RM being utilized less that 3% of the times.

To further validate our findings, we conducted an additional experiment on RewardBench data. In this setup, we used the MAB trained on Wildchat to select the most appropriate RM for each query. This selected RM was then used to choose the preferred response, which we compared against the

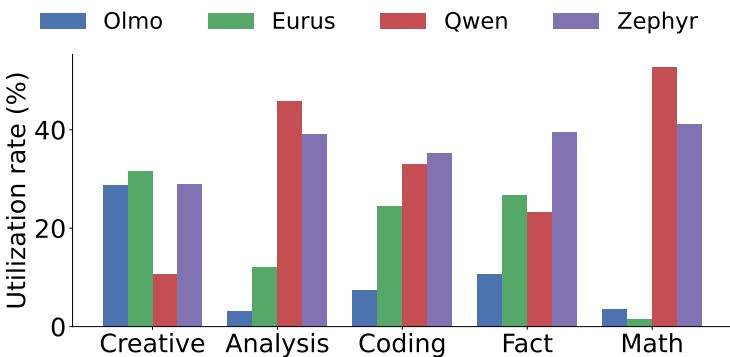

Figure 6: Utilization (%) of each RM on instruction-following queries from WildChat. The bars are arranged based on their overall scores on RewardBench, from lowest to highest. LASER dynamically selects from different RMs depending on the nature of the underlying instance.

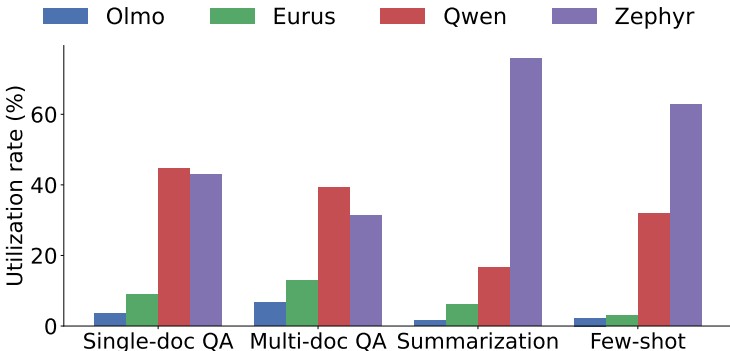

Figure 7: Utilization (%) of each RM on long-context understanding tasks. The bars are arranged based on their overall scores on LongBench, from lowest to highest. LASER dynamically selects from different RMs depending on the nature of the underlying instance.

ground-truth preferences in RewardBench. We also report the accuracy of each RM in selecting the correct preferred responses. The results are shown in Fig. 8, which demonstrates that LASER consistently selects the RM whose scoring aligns most closely with the ground-truth preferences.

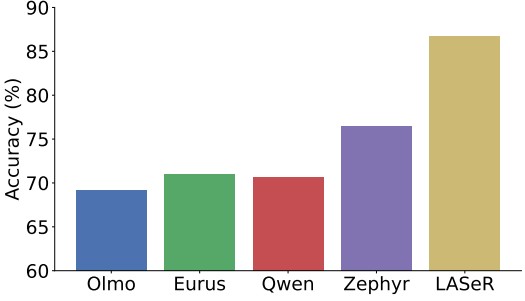

Figure 8: Accuracy (%) of each RM on selecting the preferred responses on RewardBench. LASER achieves the highest accuracy.

**MAB Rewards Stabilize Over Time.** We observed empirically that the average return (i.e., normalized MAB rewards) of each iteration tends to stabilize over time, particularly after 6-8 training iterations. Fig. 9 shows per-RM normalized MAB reward over training iterations on GSM8K. Notably, the more effective RMs (e.g., Qwen and Zephyr for math reasoning tasks) retain higher MAB rewards over time, suggesting that convergence does not imply uniformity, but rather stabilization of relative

effectiveness (this is also consistent with results in Fig. 6 and Fig. 7 that RM selection adapts to the best RM for each task).

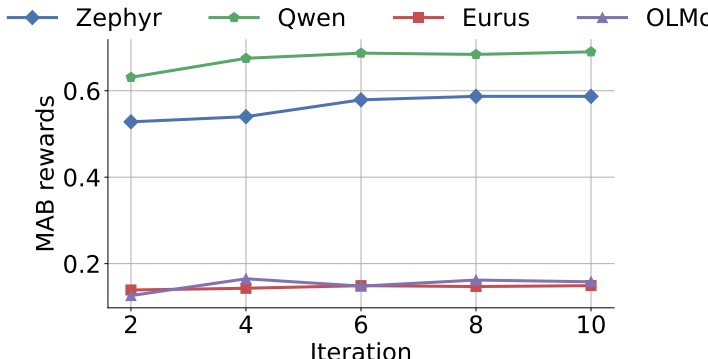

Figure 9: Average MAB rewards of different RMs over iterations on GSM-8K.

**LASER Benefits Cold-start Training and RM Adaptation.** As MAB rewards tend to converge over training iterations, the trained MAB on one dataset can be used for cold-start training on another dataset without retraining the MAB. We conducted an additional experiment where we reinitialized the LLM from the SFT checkpoint and re-trained on reasoning tasks using the converged MAB policy from a prior LASER run on Wildchat. We denote this new method as cold-start LASER (CS-LASER) and present the result in the table below. Results show a comparable performance compared to LASER (76.32% vs 76.24% on average of 3 tasks), suggesting that using a fixed, learned RM selector policy can benefit cold-start training.

Table 6: Performance of CS-LASER on reasoning tasks.

| Method | StrategyQA | GSM8K | MMLU | Avg. |
|---|---|---|---|---|
| SFT | 80.41 | 69.43 | 65.66 | 71.83 |
| LASER | **85.96** | **74.75** | 68.24 | **76.32** |
| CS-LASER | 85.78 | 74.58 | **68.37** | 76.24 |

Another benefit of LASER is that it can add or update an RM without requiring the entire MAB to be retrained from scratch. LinUCB maintains independent statistics $(A_k, b_k)$ for each RM arm. To add a new RM, we can initialize its parameters and begin exploration, allowing the bandit to learn its utility online. When adding a new RM to LASER, only the new RM's bandit parameters need to be initialized (e.g., $(A_k = I, b_k = 0)$). The existing RMs' statistics remain unchanged, meaning the MAB retains all prior learning about their utility. This allows the bandit to continue exploiting well-performing existing RMs while gradually exploring the new RM. Similarly, if an existing RM is updated (e.g., improved or retrained), the MAB can re-learn its statistics gradually through continued training. To empirically demonstrate this, we conduct an experiment where we resume training Llama-3-8B from a saved checkpoint and add a new, stronger RM QRM-Llama3-8B based on the latest RewardBench [Lambert et al., 2024] leaderboard to the current pool of RMs. We observe that the MAB quickly adapts to the new RM, with training converging after just 2 additional iterations and achieving 75.58% accuracy on the GSM8K test set, improving over LASER without QRM-Llama3-8B by 0.83%. We also run an additional experiment where we reset the MAB parameters for an existing RM (Qwen) and then resume training Llama-3-8B from a saved checkpoint. Training converges after 3 additional iterations and achieves comparable accuracy to the original checkpoint before the reset (74.72% vs. 74.75% accuracy).

Besides accuracy, we also show the utilization rates of each RM before (Checkpoint), after (Transfer) adding the new RM, and restart (Restart) in Table 7. These results show that LASER can effectively integrate new RMs and adapt its selection strategy, shifting toward using stronger RMs when they offer more informative supervision, while still retaining and utilizing previously effective RMs like Qwen for reasoning tasks. Additionally, in our restart experiment, where we reset the MAB parameters of Qwen and resume training, demonstrates that LASER can quickly relearn the utility of a strong RM, returning to similar usage levels with just a few additional iterations.

Table 7: Utilization rate (%) across RMs when including a different RM or restarting a RM.

| Utilization Rate (%) | Olmo | Eurus | Qwen | Zephyr | QRM |
|---|---|---|---|---|---|
| Checkpoint | 6.26 | 2.06 | **66.81** | 24.87 | - |
| Transfer (Add QRM) | 1.68 | 0.77 | 35.08 | 8.93 | **53.54** |
| Restart (Qwen) | 7.68 | 2.47 | **65.25** | 24.59 | - |

**Detailed Results for Each RM.** Here, we provide detailed reasoning results for each chosen RM where we use a single RM during training (c.f. Sec. 3.1) in Table 8. These results demonstrate that Qwen1.5-7B-Chat outperforms other RMs on StrategyQA and MMLU, whereas on GSM8K Zephyr-7B-Alpha has the best performance with Llama-3-8B. However, LASER still yields the best performance, outperforming all RMs by at least 1% on average across reasoning tasks, without the knowledge of which RM is most suited for each task *a priori*. This is consistent with the results on Wildchat in Table 9 where LASER outperforms all RMs with significant win rates (e.g., 63.47% and 59.02% against Qwen and Eurus, respectively).

Table 8: Performance of 4 RMs including OLMo, Eurus, Zephyr, and Qwen1.5 on Llama-3-8B. The best is bolded, and the second-best is underlined.

| Method | StrategyQA | GSM8K | MMLU | Avg. |
|---|---|---|---|---|
| OLMo-7B-Instruct | 80.23 | 68.91 | 65.74 | 71.62 |
| Eurus-7B-Kto | 81.15 | 71.13 | 66.26 | 72.84 |
| Zephyr-7B-Alpha | 84.29 | _73.16_ | 67.15 | 74.87 |
| Qwen1.5-7B-Chat | _84.79_ | 73.07 | _67.53_ | _75.13_ |
| LASER (Ours) | **85.96** | **74.75** | **68.24** | **76.32** |

Table 9: Win rates of LASER against individual RMs.

| | Win Rates |
|---|---|
| LASER vs. Zephyr | 56.34% |
| LASER vs. Qwen | _63.47%_ |
| LASER vs. Eurus | 59.02% |
| LASER vs. Olmo | **66.72%** |

**Impact of RMs Diversity.** In this experiment, we test the impact of RMs diversity on downstream performance. Instead of selecting RMs based on overall RewardBench scores across categories as in our initial experiments, we created a more capability-diverse RM set by choosing the best-performing RM in each category in RewardBench: "Chat", "Chat Hard", "Safety", and "Reasoning". This selection improves the diversity in the types of tasks and evaluation criteria each RM specializes in. The new RM set includes Qwen1.5-7B-Chat, Zephyr-7B-Alpha, Tulu-2-DPO-7B, and MMPO-Gemma-7B, replacing Eurus and OLMo models from the previous RM set. Using this updated RM set, we test LASER's ability (denoted as "LASER-Diverse") to generalize across diverse categories on WildChat. We compare LASER-Diverse to the Best RM (from the revised RM set) and RM Agreement Ensemble baselines (the strongest baselines across our benchmarks) and report the results in Table 10. These results show that with a new set of RMs, LASER continues to perform better, outperforming both the Best RM and RM Agreement Ensemble baselines. We emphasize that whether RMs are selected based on overall scores or diversity across RewardBench categories, LASER consistently outperforms baselines. This reflects the core contribution of LASER: its ability to automatically select reward models at the instance level, achieving consistent gains across tasks without requiring manual tuning or oracle selection. Users can simply provide a pool of RMs without knowing which is best.

## C   Generalization Capabilities of LASER

**Generation to other Bandit Algorithms.**

Table 10: Win rates of LASER against the two best baselines with a more diverse set of RMs.

| | Win Rates |
|---|---|
| LASER-Diverse vs. Best RM | 56.28% |
| LASER-Diverse vs. RM Agree. Ensemble | **57.21%** |

To demonstrate the generalization of LASER to other Bandit Algorithms, we include empirical comparisons with NeuralUCB, a more expressive contextual bandit and Exp3, a widely used non-contextual bandit algorithm. NeuralUCB is parameterized by a 3-layer MLP with hidden dimensions 256 and 128. The output layer of NeuralUCB produces a single scalar, which is the predicted expected reward for each RM based on the context. Table 11 shows that LinUCB achieves comparable performance to Exp3 and performs slightly below NeuralUCB on 2 out of 3 datasets. However, LinUCB is significantly more efficient and stable because it uses closed-form updates and maintains simple per-arm information, avoiding the overhead of training neural networks as in NeuralUCB. These results support our central claim: regardless of the specific bandit algorithm used, LASER consistently outperforms baseline RM selection and ensemble baselines.

Table 11: Performance and efficiency of LASER with different bandit algorithms.

| Method | StrategyQA | GSM8K | MMLU | Wall-clock hours (Avg.) |
|---|---|---|---|---|
| LASER (LinUCB) | **85.96** | 74.75 | 68.24 | **5.92** |
| LASER (Exp3) | 85.58 | 75.04 | 68.56 | 6.26 |
| LASER (NeuralUCB) | 85.73 | **75.62** | **69.15** | 10.27 |

**Generalization to Different Base Models.** In the main paper, we present the results for the Llama-3-8B model. Here we present the results for Mistral-7B model across reasoning, instruction-following and long-context understanding tasks.

In reasoning tasks, LASER improves average accuracy by roughly $2\%$ and $1.44\%$ over the sequential baseline and the best RM baseline, respectively. Compared to ensemble baselines, LASER outperforms RM Agreement Ensemble, Online RM Ensemble, and RM Score Ensemble by $1.65\%$ and $2.71\%$, respectively. Overall, we observe the same results as Llama-3-8B model, we achieve *consistent* results while the underlined second-place models show *inconsistent* performance across datasets and models.

Table 12: Performance on reasoning benchmarks on Mistral-7b. The baselines also include supervised fine-tuning on human-written responses (SFT) as a reference for performance without preference optimization. The highest accuracy is shown in bold, and the second-highest accuracy is underlined. LASER yields the highest average accuracy.

| Method | StrategyQA | GSM8K | MMLU | Avg. |
|---|---|---|---|---|
| SFT | 68.57 | 43.62 | 56.48 | 56.22 |
| Best RM | 70.06 | 45.81 | 62.04 | 59.30 |
| Avg. RM | 69.62 | 45.47 | 59.58 | 58.22 |
| Random RM Selection | 69.97 | 45.12 | 59.88 | 58.32 |
| Seq. RM Selection | 70.59 | 46.11 | 59.66 | 58.79 |
| Classifier Selection | 70.31 | 45.28 | 60.35 | 58.65 |
| RM Score Ensemble | 68.89 | 44.53 | 58.23 | 57.22 |
| RM Agree.Ensemble | 70.26 | 45.92 | 61.09 | 59.09 |
| Online RM Ensemble | 68.95 | 45.65 | 59.49 | 58.03 |
| LASER (Ours) | **73.06** | **46.89** | **62.27** | **60.74** |

In instruction-following tasks, Fig. 10 shows that LASER consistently outperforms all baselines, including Best RM (58.72%), Sequential RM Selection (63.72%), and Random RM Selection (70.61%). It achieves the highest win rate against RM Score Ensemble (73.27%) and surpasses Online RM Ensemble (65.41%), demonstrating its ability to generalize to different LLMs.

In long-context understanding tasks, Table 2 highlights LASER's superior performance on long-context understanding tasks with Mistral-7B. LASER achieves the highest scores for Single-Doc

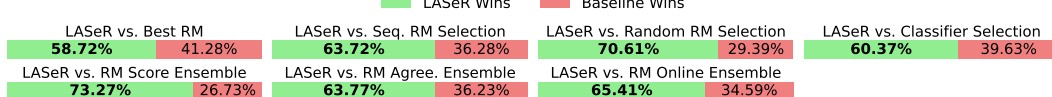

Figure 10: Length-controlled AlpacaEval win rates comparing LASER against baselines on the instruction-following tasks on WildChat using Mistral-7B.

Table 13: LASER outperforms baselines in long-context understanding tasks with Mistral-7B. Sequential RM selection is not applicable in this setting, as only inference is conducted. For QA and few-shot learning tasks, we report F1 scores, and for summarization, we report Rouge-L.

| Method | Single-Doc QA | Multi-Doc QA | Summarization | Few-shot Learning |
|---|---|---|---|---|
| Base model | 26.01 | 24.06 | 26.47 | 64.93 |
| Best RM | 28.93 | **27.93** | **30.42** | 68.34 |
| Random RM Selection | 27.44 | 25.38 | 27.19 | 66.72 |
| Classifier RM Selection | 27.59 | 25.69 | 27.97 | 67.04 |
| RM Score Ensemble | 26.75 | 25.71 | 28.17 | 66.97 |
| RM Agree. Emsemble | 27.96 | 26.60 | 28.27 | 67.23 |
| Online RM Ensemble | 27.55 | 25.42 | 28.63 | 66.81 |
| LASER (Ours) | **29.14** | 27.80 | 30.08 | **68.47** |

QA (29.14) and Multi-Doc QA (27.80), significantly outperforming the base model. In Few-Shot Learning, LASER achieves 68.47, surpassing both the Best RM (68.34) and Online RM Ensemble (66.81). While its Rouge-L score for summarization (30.08) is slightly below Best RM (30.42), LASER remains comparable. These results demonstrate the consistent performance of LASER across the model family.

**Generalization to the Base and RMs of Bigger Sizes.** To further analyze the generalization of LASER to the base and RMs of bigger sizes, we add an experiment with Qwen2.5-32B [Qwen Team, 2024a] policy model and the following 4 similarly sized RMs from RewardBench: RISE-Judge-Qwen2.5-32B, Bagel-DPO-34B, Archangel-DPO-30B, Archangel-KTO-30B. We compare LASER with the best RM and the RM Agreement Ensemble baselines, which are the second-best baselines for most of the benchmarks on Wildchat. Table 14 demonstrates that LASER has higher win rates than baselines.

Table 14: Win rates of LASER compared to Best RM and RM Agreement Ensemble baselines on Qwen-32B model.

| | Win Rates |
|---|---|
| LASER vs. Best RM | **54.44%** |
| LASER vs. RM Agree. Ensemble | **55.78%** |

**Generalization to Correctness Rewards for Reasoning.** As verifiable rewards emerge as a powerful RM for reasoning tasks, in this experiment, we analyze how does it compared with current multiple RMs framework of LASER, and whether it is possible to incorporate them into LASER.

First, we implement a Correctness Reward baseline for the reasoning datasets. Specifically, we assign a reward of 1 if the generated answer matches the correct final answer and 0 otherwise. We maintain the same iterative training setup as in our main experiments and compare with LASER training with multiple RMs. Table 15 shows that while the correctness-based reward performs competitively, especially on GSM8K (74.56% vs. 74.75% accuracy), LASER outperforms it across all tasks, including commonsense reasoning tasks (StrategyQA and MMLU). This is likely because correctness-based rewards are well-suited for tasks with clearly verifiable outputs, such as math or code, but they struggle to capture the multi-step reasoning and contextual understanding needed in factual or open-domain QA, where answers are often ambiguous or depend on background knowledge. In these cases, other forms of supervision such as process-based rewards [Ma et al., 2025] or human preference signals modeled by RMs are typically more effective. LASER is designed to leverage this by selecting among multiple RMs, allowing it to perform well across a variety of task types.

Table 15: Performance comparison of LASER and the Correctness Reward baseline.

| Method | StrategyQA | GSM8K | MMLU | Avg. |
|---|---|---|---|---|
| Correctness Reward | 83.42 | 74.56 | 66.35 | 74.78 |
| LASER | **85.96** | **74.75** | **68.24** | **76.32** |

As noted in Sec. 3, LASER can treat any RMs as part of its selection framework. We show that LASER can integrate correctness-based rewards, combining their strengths with those of preference-based models. To validate that, we run an additional experiment on GSM8K, where we add the correctness-based RM to the existing pool of learned RMs (Olmo, Eurus, Qwen, Zephyr). To further test LASER's ability to adaptively select appropriate RMs, we create a mixed dataset combining GSM8K with WildChat (subsampled to 1.2K examples per category to ensure balance). Since correctness-based signals are not applicable to WildChat's open-ended prompts, we assign a random binary reward (0 or 1) for those examples to simulate a misleading RM. We observe that LASER achieves an accuracy of 75.06% accuracy on the GSM8K test set, improving over LASER without correctness rewards by 0.31%. We also report the utilization rate of each RM in the pool (Olmo, Eurus, Qwen, Zephyr, correctness/random) for two types of queries: GSM8K and WildChat in Table 16. On GSM8K, LASER heavily leverages the correctness-based RM (40.26%) alongside strong learned RMs like Qwen (43.58%), indicating that it effectively identifies and uses the most reliable supervision. In contrast, for WildChat, LASER downweights the misleading correctness-based RM (2.84%) and shifts toward more suitable RMs like Zephyr and Qwen. These results demonstrate LASER's ability to adapt RM usage based on task domain, even in the presence of noisy or misleading supervision.

Table 16: Utilization rate (%) of different RMs including a correctness RM and a random RM on GSM8K and WildChat.

| Utilization Rate (%) | Olmo | Eurus | Qwen | Zephyr | Correctness/Random |
|---|---|---|---|---|---|
| GSM8K | 4.02 | 2.32 | 43.58 | 19.82 | 40.26 |
| WildChat | 9.09 | 16.41 | 27.75 | 45.90 | 2.84 |

**Generalization to the Number of RMs.** To study the generalization capability of LASER across the number of RMs, we expand the candidate set of RMs with up to 4 more RMs from the RewardBench leaderboard, including Tulu-2-DPO-7B [Ivison et al., 2023], Zephyr-7B-Gemma [Tunstall and Schmid, 2024], Qwen1.5-MoE-A2.7B-Chat [Qwen Team, 2024b], Archangel-7B [Ethayarajh et al., 2024]. Fig. 11 shows that the accuracy remains consistent across all datasets as the number of RMs varies. StrategyQA remains near 85.9%, GSM8K around 74.8%, and MMLU close to 68.1%, with minimal fluctuations, indicating robust performance regardless of the number of RMs while RM Agreement Ensemble performance declines more steeply.

**Generalization to Training Loss Functions.** In Sec. 3.1, we state that the choice of loss function used to train the LLM depends on the underlying task or domain. Nevertheless, we always use the training loss as the MAB reward to update the MAB's parameters. Here we study the performance of LASER and baselines with 4 different loss functions, NLL, DPO, NLL + DPO [Pang et al., 2024], and KTO [Ethayarajh et al., 2024], in the reasoning domain. Results in Table 17 show that training LLMs with multiple rewards using LASER outperforms sequential RM selection by 2.4%, 1.7%, and 1.3% when using NLL, DPO, NLL+DPO loss functions, respectively; while both methods yield comparable performance with KTO. Additionally, we found that the most effective training loss functions are NLL + DPO for StrategyQA, NLL for GSM8K, and KTO for MMLU. However, irrespective of the choice of the underlying loss function, LASER is more effective at balancing and adaptively selecting from multiple RMs. Lastly, we compare LASER with a variant in which we use $\text{Acc}(y^w) - \text{Acc}(y^l)$ as the MAB reward, which uses the ground-truth information about the final answer. We find that using the negative training loss of the LLM is more effective than using accuracy as the MAB reward.

**Generalization to OOD Tasks.** We first assess the generalization ability of our method by training models on specific datasets and evaluating their performance on out-of-distribution reasoning tasks. Specifically, we train the model on the StrategyQA and MMLU datasets and evaluate its generalization on the CommonsenseQA [CSQA; Talmor et al., 2019] dataset. Similarly, we train on GSM8K and test on MATH [Hendrycks et al., 2021c] to assess the model's ability to generalize across different

Table 17: Across different training loss functions, optimizing with multiple RMs via LASER outperforms the sequential RM selection with Llama-3-8B. SQA denotes StrategyQA.

| Loss | Method | SQA | GSM8K | MMLU | Avg. |
|---|---|---|---|---|---|
| NLL | Sequential | 82.75 | 71.80 | 65.41 | 73.32 |
|  | LASER | 85.11 | 74.94 | 67.09 | **75.71** |
| DPO | Sequential | 83.26 | 71.94 | 65.38 | 73.53 |
|  | LASER | 84.71 | 73.94 | 67.02 | **75.22** |
| KTO | Sequential | 83.62 | 73.07 | 69.02 | 75.24 |
|  | LASER | 84.87 | 73.86 | 69.05 | **75.66** |
| NLL+DPO | Sequential | 83.90 | 72.94 | 68.02 | 74.95 |
|  | LASER w/. Acc | 83.04 | 73.12 | 65.46 | 73.87 |
|  | LASER | 85.96 | 74.75 | 68.24 | **76.24** |

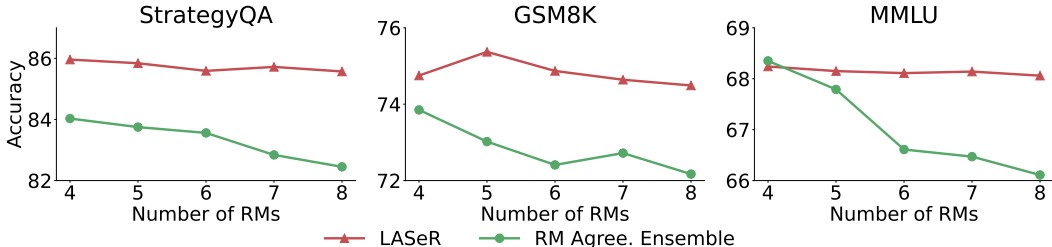

Figure 11: LASER's performance is robust to adding weaker RMs to the candidate set to select from.

reasoning datasets. These tasks are designed to capture both general reasoning ability and OOD generalization across domains. We report the results in Table 18, where we find that across both Llama-3-8B and Mistral-7B models, models trained with LASER yield the best average accuracy, beating training with the best RM by roughly $2\%$ (absolute) on CSQA with Llama-3-8B. On Mistral-7B, training with LASER outperforms both training with single best RM and sequential RM selection by slightly over $1\%$.

Table 18: Generalization performance of different models trained on StrategyQA, MMLU, and GSM8K, and evaluated on CSQA and MATH, respectively.

| Method | CSQA | MATH | Avg. |
|---|---|---|---|
| **Llama-3-8B** | | | |
| SFT | 65.64 | 29.13 | 47.39 |
| Best RM | 67.59 | 31.54 | 49.57 |
| Avg. RM | 67.16 | 30.36 | 48.76 |
| Random RM Selection | 68.31 | 30.21 | 49.26 |
| Sequential RM Selection | 67.73 | 30.25 | 48.99 |
| **LASER (Ours)** | **69.26** | **31.67** | **50.47** |
| **Mistral-7B** | | | |
| SFT | 59.06 | 16.38 | 37.72 |
| Best RM | 60.46 | 18.08 | 39.27 |
| Avg. RM | 60.06 | 17.25 | 38.65 |
| Random RM Selection | 60.61 | 16.96 | 38.58 |
| Sequential RM Selection | 60.56 | 17.96 | 39.26 |
| **LASER (Ours)** | **61.65** | **18.97** | **40.31** |

**Robustness to Noisy Rewards.** To examine the robustness of our method in the presence of noisy or irrelevant rewards, we conduct the following analysis using Llama-3-8B on GSM8K. We add varying amounts of Gaussian noise $\sigma$ to the rewards generated by RMs sampled from the distribution $\mathcal{N}(0, \sigma I)$, where $I$ is the identity matrix, to simulate noisy rewards when using RMs in out-of-

distribution settings. In addition to LASᴇR using the LinUCB algorithm (c.f. Sec. 3.2), we also use Exp3 [Auer et al., 2002] designed for adversarial bandit settings. In Fig. 12, we find that even as the degree of noise in RM scores increases (from $\sigma = 0.1$ to 0.4), LASᴇR's selection strategy continues to perform robustly, mitigating the effects of noise compared to the sequential baseline. Specifically, LASᴇR has an average performance drop of only 0.55% accuracy at a noise level of $\sigma = 0.3$, whereas the sequential baseline suffers a 1.6% accuracy drop (3 times as much) under the same conditions. Furthermore, LASᴇR using Exp3, the most noise-robust method, maintains consistent performance, with only a 0.26% accuracy drop.

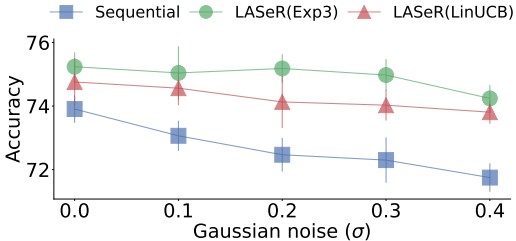

Figure 12: Impact of the magnitude of Gaussian noise on the accuracy of LASᴇR and sequential baseline on RewardBench.

**LASᴇR Training with Domain-specific Evaluation Metrics.** While recent works focus on building RMs that reflect preferences across domains, an extensive body of prior work develops a suite of evaluation metrics catered to specific domains such as reasoning [Golovneva et al., 2022, Prasad et al., 2023]. To show that LASᴇR can be used to select any kind of evaluation metric from a collection of metrics during training, in Table 19, we present results with training LLMs on model-based metrics from ROSCOE [Golovneva et al., 2022] by replacing RMs with informativeness, faithfulness, reasoning alignment, hallucination, common sense error, semantic, coherence and perplexity in Sec. 3. Llama-3-8B models trained using LASᴇR yield 1.62% accuracy improvement over baselines on average across three datasets. These results are also generalized to Mistral-7B, except for GSM8K, where we achieve comparable performance to the Base + Informativeness baselines. Note that the perplexity of most responses is nearly identical, making it difficult to distinguish between them, explaining why perplexity shows little to no improvement compared to the base model.

Table 19: Comparison of LASᴇR and baselines on ROSCOE. The baselines include supervised fine-tuning (SFT), sequential optimization, uniform rewards selection, and base model optimized with one specific evaluation metric (Perplexity, Informativeness).

| Method | StrategyQA | GSM8K | MMLU | Avg. |
|---|---|---|---|---|
| **Llama-3-8B** | | | | |
| SFT | 80.41 | 69.43 | 65.66 | 71.83 |
| Perplexity | 80.55 | 69.21 | 65.62 | 71.79 |
| Informativeness | 82.87 | 73.55 | 66.69 | 74.37 |
| Random RM Selection | 82.72 | 70.93 | 66.10 | 73.25 |
| Seq. RM Selection | 83.15 | 73.38 | 66.17 | 74.23 |
| **LASᴇR (Ours)** | **83.54** | **73.80** | **66.79** | **74.71** |
| **Mistral-7B** | | | | |
| SFT | 68.57 | 43.62 | 56.48 | 56.22 |
| Perplexity | 68.83 | 43.47 | 57.14 | 56.48 |
| Informativeness | 70.29 | **44.98** | 59.29 | 58.19 |
| Random RM Selection | 69.24 | 44.05 | 57.68 | 56.99 |
| Seq. RM Selection | 70.40 | 44.79 | 59.07 | 58.09 |
| **LASᴇR (Ours)** | **70.91** | 44.93 | **59.63** | **58.49** |

**Robustness to Underperforming Evaluation Metrics.** Similar to our analysis on noise in rewards in Sec. 5, we investigate how adding ROSCOE metrics with poor correlation to human-annotated labels in meta-evaluation by Golovneva et al. [2022] impacts the performance of Llama-3-8B on GSM8K. Once again, even with ROSCOE metrics, demonstrates LASᴇR can maintain consistent

performance by adaptively prioritizing the most relevant reward signals, outperforming the sequential baseline, which fails to filter out irrelevant information effectively. Fig. 13 shows that as the number of irrelevant metrics increases, LASeR's selection strategy continues to perform robustly. Specifically, LASeR has an average performance drop of only 0.13%, whereas the sequential baseline suffers a 2.15% accuracy drop under the same conditions. Lastly, LASeR using Exp3 maintains a consistent performance level with a 0.4% accuracy drop.

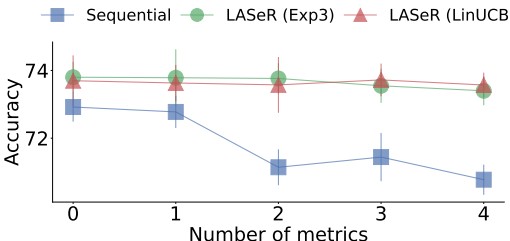

Figure 13: Impact of irrelevant metrics from ROSCOE on the GSM8K accuracy of LASeR and sequential baseline.

## D  Ablation Study

**Context Embedding.**  As noted in Sec. 3, to validate the effectiveness of the last-token embedding method, we conduct an ablation study comparing three context embedding approaches: first-token (using the embedding of the first token in the prompt), average-token (using the average of all token embeddings in the prompt), and last-token (using the embedding of the last token in the prompt). We evaluate these methods on Llama-3-8B and reasoning tasks. Table 20 shows that the last-token embeddings consistently outperform the alternatives across all three datasets, demonstrating the effectiveness of this method in LASeR.

Table 20: Comparing different token embedding methods in LASeR.

| Method | StrategyQA | GSM8K | MMLU | Avg. |
|---|---|---|---|---|
| LASeR (first) | 83.59 | 73.03 | 67.03 | 74.55 |
| LASeR (avg.) | 83.22 | 73.47 | 66.52 | 74.40 |
| LASeR (last) | **85.96** | **74.75** | **68.24** | **76.32** |

**Impact of Bandit Parameter $\alpha$.**  We conduct an ablation study to assess the impact of the exploration parameter $\alpha$ in the LinUCB algorithm on Llama-3-8B and reasoning tasks. Table 21 shows that for reasoning datasets, disabling exploration leads to suboptimal performance, particularly in early training stages where the bandit may overfit to a suboptimal RM. In contrast, enabling exploration ($\alpha > 0$) improves performance by allowing the model to discover and leverage more informative RMs over time. We find that setting $\alpha$ in the range of 0.4 to 0.6 consistently yields the best results across tasks, offering a good balance between exploration and exploitation. This is consistent with our observations on the performance on dev set for reasoning tasks as noted in Appendix A.1.

Table 21: Effect of varying $\alpha$ on accuracy for reasoning tasks.

| Dataset | $\alpha$=0.0 | $\alpha$=0.2 | $\alpha$=0.4 | $\alpha$=0.6 | $\alpha$=0.8 | $\alpha$=1.0 |
|---|---|---|---|---|---|---|
| StrategyQA | 83.11 | 85.29 | **85.96** | 85.32 | 84.53 | 83.77 |
| GSM8K | 71.68 | 73.22 | 74.10 | **74.75** | 74.63 | 73.94 |

## E  Additional Discussion

**LASeR with different "kinds" of RMs.**  In Sec. 4.2, we show that LASeR can choose from a set of candidate RMs, and our analysis in Fig. 12 highlights the fact that LASeR is robust to noisy

RMs. In Appendix C, we show that LASER can also be used with metric-based rewards [Golovneva et al., 2022]. These experiments reflect a conceptual split between the generator (the LLM) and the scorer (the RM or metric). Thus, LASER is applicable to other settings that follow this paradigm, e.g., using an LLM-as-a-judge [Zheng et al., 2023], where LASER could be used to choose between different judge models, prompts, or different combinations of RMs and metrics. However, consistent with the "self-preference" bias of LLMs [Panickssery et al., 2024], we caution that using an RM that is based on the same model as the LLM used for generating responses could lead to the MAB spuriously favoring certain RMs. We leave further study on extending LASER to future work.

**Quality of RMs used with LASER.** Methods that rely on RMs for scoring generally assume that these RMs have a strong correlation with human judgments. LASER tempers this assumption in a number of ways: First, by ensembling multiple RMs, LASER weakens the effect of noisy RMs; this can be seen in Fig. 12, where LASER mitigates the negative impact of a noisy RM even as the level of noise is increased. Moreover, the fact that LASER can select RMs at an instance level means that there need not be a *single* RM that always correlates well across all instances. However, LASER does require at least one RM to be positively correlated with human judgments on each instance. If this assumption is not met (i.e., *all* RMs are poorly correlated across *all* instances), then optimizing for the RMs will yield poor results. Note that this holds true for any method optimizing for RMs. Because LASER selects from multiple RMs, its contributions are complementary to developments in RMs, which can easily be integrated into LASER, as well as improvements to preference optimization loss functions (see Appendix C). Such RM improvements are likely to be necessary as LLMs are deployed in domains that are out of scope for existing systems and domains with heterogeneous requirements (e.g., our generation domains in Sec. 4.2). In these cases, there will be no single "perfect" existing RM, and successful solutions will likely involve mixing multiple RMs. A core benefit of LASER is its ability to automatically filter RMs; in Fig. 6 we see that utilization differs across domains. This allows users to avoid expensive experimentation with subsets of RMs: they can simply offload this task to LASER, which will automatically select the more useful RM(s).

# F    Prompts

We provide the prompts used for our experiments as follows:

---

**Reasoning**

**Prompt:** Your task is to answer the question below. Give step by step reasoning before you answer, and when you're ready to answer, please use the format "Final answer:..."
**Question:** {input}
**Solution:**

---

**Instruction-Following**

**Prompt:** You are an assistant capable of assisting users in various tasks, including creative writing, analysis of texts and data, coding, providing factual information, and solving math problems. For creative writing, help users brainstorm ideas and develop their narratives. For analysis, guide users in breaking down concepts and exploring different perspectives. In coding, assist with programming questions and debugging. When providing factual information, ensure accuracy and cite reliable sources. For math reasoning, offer step-by-step solutions and encourage logical thinking. Maintain a clear, engaging, and supportive tone throughout your responses to foster learning and creativity.
**Question**: {input}
**Answer:**

---

## Long-Context Understanding

**Single-Doc QA:**
**Prompt:** You are given a scientific article and a question. Answer the question as concisely as you can, using a single phrase or sentence if possible. If the question cannot be answered based on the information in the article, write "unanswerable". If the question is a yes/no question, answer "yes", "no", or "unanswerable". Do not provide any explanation.
Article: context Answer the question based on the above article as concisely as you can, using a single phrase or sentence if possible. If the question cannot be answered based on the information in the article, write "unanswerable". If the question is a yes/no question, answer "yes", "no", or "unanswerable". Do not provide any explanation.
**Question:** {input}
**Answer:**

**Multi-Doc QA:**
**Prompt:** Answer the question based on the given passages. Only give me the answer and do not output any other words. The following are given passages.
{context}
Answer the question based on the given passages. Only give me the answer and do not output any other words.
**Question:** {input}
**Answer:**

**Summarization:**
**Prompt:** You are given several news passages. Write a one-page summary of all news.
News: {context}
Now, write a one-page summary of all the news.
**Summary:**

**Few-shot Learning:**
**Prompt:** Answer the question based on the given passage. Only give me the answer and do not output any other words. The following are some examples.
{context}
**Question:** {input}
**Answer:**

