# OpenReview forum: "LASeR: Learning to Adaptively Select Reward Models with Multi-Arm Bandits"
_NeurIPS.cc/2025/Conference — NeurIPS 2025 poster_

### Official Review · Reviewer_PKfF · 2025-06-10

**Clarity:** 4
**Significance:** 3
**Originality:** 3
**Rating:** 5
**Confidence:** 4

**Summary:**

When finetuning LLMs to align with human preferences (RLHF), it is unclear how to pick the most suitable reward model for a particular domain. Finetuning with multiple reward models at once can be expensive (due to multiple forward passes) and noisy (due to conflicting rewards).

This paper proposes to incorporate a contextual bandit model (LinUCB) into the RLHF process. The idea is to adaptively pick a _single_, most suitable reward model to score each prompt-answer pair.

Experiments suggest that this approach is effective in making RLHF both more cost-efficient and effective.

**Questions:**

Could the authors please address the "Weaknesses" list above?

**Ethical Concerns:**

["NO or VERY MINOR ethics concerns only"]

**Final Justification:**

During the rebuttal, the authors addressed all of my concerns sufficiently, especially regarding the positioning of this paper and the significance of the results. I highly urge the authors to put all the explanations provided during the rebuttal phase in the main text of the manuscript.

**Limitations:**

Not included in the main text. I suggest the authors fix this and be more upfront about the limitations.

**Paper Formatting Concerns:**

Minor, see above.

**Quality:**

3

**Strengths And Weaknesses:**

**Strengths**

1. Very clear and well-written paper
2. Tackling a relevant, real-world problem
3. Extensive experimental validation
4. Straightforward method---a strength since it is easy to implement


**Weaknesses**

1. Limited novelty: The proposed method is essentially just a straightforward application of LinUCB.
2. Lack of theoretical analysis. I believe that the previous point is fine if the authors provide some theoretical analyses for this specific use case. For example, the validity of using the negative log-likelihood as a reward (e.g. the LLM might be misspecified, so this reward might be biased, or easy to "hack", etc.).
3. It is unclear whether the results are significant. The numbers are very close together, and not even error bars are provided. I suggest the authors provide some non-parametric hypothesis tests on the results.

---

> ### Author Rebuttal · Authors · 2025-07-30
>
> We thank you for your review and suggestions. We appreciate your recognition of our paper as “well-written”, our efforts to address “a relevant, real-world” problem, and conduct “extensive” experiments. Below, we respond to your comments and suggestions in detail.
>
> ---
>
> > Novelty of LASeR (beyond using LinUCB)
>
> We would like to clarify that the primary contribution of our work is the **formulation of preference-based LLM training as a MAB problem over multiple RMs, a novel and previously unexplored setting**. This problem is challenging because RM quality can vary across tasks and examples, yet most existing methods assume a fixed RM throughout training. Our key insight is to treat RM selection as an instance-level, contextual MAB problem, allowing the model to adaptively choose the most informative RM for each input. This framing enables us to integrate RM feedback directly into the bandit loop during preference fine-tuning, rather than using MABs only for data selection (Dwaracherla et al., 2024) or model selection at inference time without RMs (Nguyen et al., 2024; Li, 2025). To our knowledge, LASeR is the first method to do that in the context of aligning LLMs with multiple RMs.
>
> As discussed in lines 207–210, our focus is **not** on proposing a new MAB algorithm. Moreover, the effectiveness of LASeR does not rely on LinUCB specifically, but rather on the exploration-exploitation trade-off enabled by the MAB framework. LASeR is agnostic to the specific choice of bandit algorithm (see Section 3, lines 207-210), and we demonstrate this empirically in Appendix C (lines 1225-1234, Table 9) by comparing LinUCB with alternative MAB methods (Exp3 and NeuralUCB). The consistent performance across algorithms supports our claim that the contribution lies in the integration of MABs into the LLM training pipeline.
>
> ---
>
> > Theoretical analysis
>
> We acknowledge that theoretical analysis is a valuable direction for future work. However, developing such analysis in our setting falls outside the primary focus of this work. Theoretical modeling is complicated for this setting by several factors: the non-stationary dynamics of LLM training, the non-convex nature of the optimization landscape, and the use of reward signals from imperfect and potentially biased RMs. These challenges are shared across recent work in preference-based LLM alignment (e.g., Pang et al., 2024; Dong et al., 2025; Ye et al., 2025), which also focus on empirical performance rather than theoretical guarantees. While we recognize that RMs may be biased or misspecified, LASeR mitigates the risk of reward hacking by explicitly avoiding optimization against any single fixed RM.
>
> Moreover, we believe our use of the negative DPO loss as the MAB reward is well-justified for the following reasons:
> * **Preference modeling**: The DPO loss is aligned with the preference model in common RLHF setups and reflects how clearly the LLM distinguishes preferred from rejected responses. A lower DPO loss corresponds to higher preference margin, making it a reasonable proxy for reward quality. In our framework, an RM that provides more informative and consistent rankings is given a higher reward compared to a less-suited RM providing uninformative rankings (as discussed in Section 3, lines 180-187).
> * **No more susceptible to reward hacking than optimizing RMs directly**: Unlike settings where a single RM is directly optimized (e.g., via RL or DPO with a fixed RM), LASeR avoids such issues by never optimizing against any one RM. Instead, it adaptively evaluates and switches between multiple RMs, reducing the risk of overfitting to a misspecified or hackable reward (as discussed in Section 2, lines 102-117).
> * **Empirical results**: To support the validity of our MAB reward reward, we empirically compare our approach (using negative training loss as the MAB reward) with a variant that uses accuracy difference ($\text{Acc}(y_w) - \text{Acc}(y_l)$), which relies on ground-truth answers. As shown in Appendix C (lines 1280–1283, Table 13), using negative training loss results in stronger performance gains, suggesting it serves as a reliable proxy for RM signal quality.
>
> ---
>
> > I suggest the authors provide some non-parametric hypothesis tests on the results.
>
> We note that we included an analysis in Appendix B (lines 1146–1156), where we report results over 5 random seeds to demonstrate the robustness of LASeR across datasets. To further address the reviewer’s suggestion, we extended our experiments to 10 random seeds and conducted non-parametric hypothesis testing on the reasoning benchmarks. Specifically, we test the following:
> * Null hypothesis ($\mathcal H_0$): LASeR’s accuracy is equal to the competing method’s accuracy.
>
>
> * Alternative hypothesis ($\mathcal H_a$): LASeR’s accuracy is greater than the competing method’s accuracy.
>
> We use the non-parametric one-sided Wilcoxon signed-rank test to paired accuracy values across seeds. The p-value of the test between LASeR and the two best baselines is reported in the following table. At a significant level of 0.05, the table indicates that we should reject the null hypothesis, which shows that LASeR significantly outperforms both the strongest single RM and the most competitive ensemble baseline across all three tasks.
>
> | LASeR vs. Baseline (p-value) | StrategyQA    | GSM8K    | MMLU    |
> |----------------|------|------|------|
> | LASeR vs. Best RM | 0.036 | 7e-4 | 0.019 |
> | LASeR vs. RM Agree. Ensemble  | 8e-3 | 0.024 |  0.046 |
>
>
> We hope our response has addressed all of your questions and will allow you to revisit your score.
>
> ---
>
>
> **References**
>
>
> Dwaracherla, Vikranth, et al. "Efficient exploration for llms." ICML 2024.
>
> Nguyen, Quang H., et al. "Metallm: A high-performant and cost-efficient dynamic framework for wrapping llms." arXiv preprint arXiv:2407.10834 (2024).
>
> Li, Yang. “LLM Bandit: Cost-Efficient LLM Generation via Preference-Conditioned Dynamic Routing.” arXiv preprint arXiv:2502.02743 (2025).
>
> Pang, Richard Yuanzhe, et al. "Iterative reasoning preference optimization." NeurIPS 2024.
>
> Dong, Qingxiu, et al. "Self-boosting large language models with synthetic preference data." ICLR 2025.
>
> Ye, Ziyi, et al. "Learning LLM-as-a-judge for preference alignment." ICLR 2025.

---

> > ### Comment · Reviewer_PKfF · 2025-08-04
> >
> > Thanks a lot for the rebuttal and running the suggested hypothesis testing.
> >
> > I do have one follow-up discussion point (ellipsis mine):
> >
> > > [...] We would like to clarify that the primary contribution of our work is the formulation of preference-based LLM training as a MAB problem over multiple RMs, a novel and previously unexplored setting. [...]
> >
> > This is indeed my original criticism of the paper---I didn't mean that you should've proposed a novel bandit algorithm. That is: the effectiveness of contextual MAB is well-known already, and it is a very general algorithm. So, it is not surprising that cMAB will do well in this online, reward-model selection problem. Hence, the limited novelty I mentioned above.
> >
> > This makes the paper's story sound like "yes, we can apply cMAB on this LLM setting", which is not surprising. Which is why I mentioned that this is completely fine if the main focus is on theoretical analyses. But since it's not, then there should be some other _surprising_ focus/angle that the authors tackle. And I'm not sure what this angle is at the moment.

---

> > > ### Author Response · Authors · 2025-08-05
> > >
> > > Thank you for the follow-up and for participating in the discussion. While we agree that contextual MAB is well-established, we would like to highlight several reasons why applying MABs to preference-based LLM training over multiple RMs is **not** a direct or trivial extension:
> > >
> > > * **Training dynamics**: In typical MAB problems, the arms are often fixed with stable reward distributions. In contrast, during preference fine-tuning of LLMs, the arms are RMs whose outputs interact with and influence the training of the LLM itself. As the LLM is updated, the distribution of responses (and therefore the RM-derived reward signal) shifts. LASeR uses **contextual information from the LLM** at each training step to estimate which RM will be most beneficial at a specific iteration, allowing the MAB to adapt dynamically.
> > > * **MAB reward**: Unlike traditional MAB settings where rewards are observed directly, in our setting, no explicit supervision is available to indicate which RM is best per query. In LASeR the MAB receives a **proxy reward signal** derived from the negative DPO loss, which we justified as effective based on both intuition (Section 3, lines 180-187) and empirical results (Appendix C - lines 1280–1283, Table 13).
> > > * **Adaptivity to new or noisy RMs**:  A static MAB setup struggles when new RMs are introduced or when existing RMs are noisy or domain-specific. LASeR goes beyond offline evaluation and demonstrates that it can adapt online to new, noisy, or partially applicable RMs. It is designed to **treat any reward models, including learned RM signals or domain-specific metrics as an RM** (Appendix C, lines 1307-1319) within its selection framework. This is not feasible in standard MAB setups where the model receives direct scalar rewards from RMs, as differences in reward scale would bias the bandit selection unfairly.
> > >
> > > In summary, while MAB is well-known, **its integration into preference-based LLM training introduces unique challenges not captured by standard formulations and offers surprising challenges**. LASeR addresses these with a practical and extensible framework, a contribution we believe goes beyond a direct application of existing algorithms.
> > >
> > > We would also like to politely point to the **NeurIPS review guidelines** on originality, which notes: “Does this work offer a novel combination of existing techniques, and is the reasoning behind this combination well-articulated?” and “Originality does not necessarily require introducing an entirely new method. Rather, a work that provides novel insights by evaluating existing methods, or demonstrates improved efficiency, fairness, etc. is also equally valuable.” We believe our paper fits this criterion: we introduce a novel combination between MAB and iterative preference-based LLM training, and show it leads to strong empirical gains across tasks.
> > >
> > > We hope this clarifies your concerns and will allow you to reconsider your evaluations.

---

> > > > ### Comment · Reviewer_PKfF · 2025-08-05
> > > >
> > > > Thank you for elaborating.
> > > >
> > > > Indeed, I am well aware of that review guideline. Please see my inquiries as an effort to help you articulate the novelty/surprise angle of the paper better, rather than as being adversarial.
> > > >
> > > > In the revised version, I urge the authors to better clarify various aspects of the paper as suggested here and also in other reviewers' comments.
> > > >
> > > > I don't have further questions and will provide my final acknowledgment after discussing with other reviewers and the AC.

---

> > > > > ### Author Response · Authors · 2025-08-05
> > > > >
> > > > > Thank you for actively participating in the discussion. We absolutely did not interpret your comments as adversarial. Our reference to the review guideline was simply intended to highlight how we view the contribution of our framework. We really appreciate your constructive feedback in helping us improve the paper. We will incorporate your suggestions along with those from the other reviewers into the revised version.

---

### Official Review · Reviewer_3mJq · 2025-06-27

**Clarity:** 3
**Significance:** 2
**Originality:** 2
**Rating:** 4
**Confidence:** 4

**Summary:**

The paper introduces LASER, an adaptive way to fine-tune large language models. Rather than relying on a single reward model or a static group of them, LASER treats the selection as a dynamic choice. For each new batch of prompts, it picks the single most suitable reward model for the task. This method avoids mixed signals of running multiple models simultaneously, yet still captures the distinct advantages of each one.

**Questions:**

From my understanding, the authors aim to address the meta-level question of selecting the appropriate RM depending on the training stage of the model.

- **Concern 1:** It's unclear to me how the last-token embedding alone sufficiently encodes the necessary information to predict which RM to use. Could you empirically elaborate on why this is the case?

- **Concern 2:** My other concern relates to understanding why the method works effectively. As I understand it, only one RM is selected per batch, but batches typically contain diverse samples with varying difficulties and type of problems. Consequently, averaging embeddings from multiple samples could produce a noisy estimate of context. Could the authors clarify or provide insights into why their method succeeds despite this issue?

- **Concern 3:** I believe additional ablation studies could help clarify why the MAB approach works, addressing the concerns mentioned above. For instance, how does the accumulated selection frequency for each RM behave—does it change abruptly, or is there a gradual transition between RMs as certain abilities are learned?

- **Concern 4:** Additionally, could you perform an ablation study on the parameter $\alpha$? For example, setting $\alpha = 0$ represents greedy selection; does introducing exploration ($\alpha > 0$) significantly impact performance?

**Ethical Concerns:**

["NO or VERY MINOR ethics concerns only"]

**Final Justification:**

Author have addressed my concerns

**Limitations:**

Yes

**Paper Formatting Concerns:**

.

**Quality:**

3

**Strengths And Weaknesses:**

### Strengths:
- The authors tackle an interesting problem, essentially addressing the high-level question, "which model should serve as the teacher/RM at a given iteration," if I understand correctly.
- Their framing of the problem as a contextual multi-armed bandit (MAB) makes sense.

### Weaknesses:
- My main concern, detailed below (in questions), is that while using a multi-armed bandit approach is intuitive, the current explanation isn't sufficient to fully justify why the proposed method works.

---

> ### Author Rebuttal · Authors · 2025-07-30
>
> We thank you for your detailed review and for acknowledging the “interesting” problem we solve and that the problem framing “makes sense”. Please find our responses to your comments below.
>
> ---
>
> > From my understanding, the authors aim to address the meta-level question of selecting the appropriate RM depending on the training stage of the model.
>
> To clarify, LASeR *does not solely* select RMs based on the training stage of the LLM. Rather, our method performs instance-level RM selection using a contextual MAB that takes into account both the input prompt (via its embedding) and the current training dynamics (via the model’s evolving response quality and loss) (as discussed in Section 3, lines 195-206). We also showed that LASeR can be applied to training-free settings such as long-context understanding tasks for reranking responses (Section 4, lines 384-370).
>
> ---
>
> > 1. It's unclear to me how the last-token embedding alone sufficiently encodes the necessary information to predict which RM to use. Could you empirically elaborate on why this is the case?
>
> We use the last-token embedding as the context representation because in Transformer-based LLMs, this position typically aggregates information from the entire sequence. Moreover, this method is a standard and commonly used approach for extracting LLM embeddings (Wang et al., 2024,  BehnamGhader et al., 2024). To empirically validate this choice, we conducted an ablation comparing three context embedding methods: first-token, average-token, and last-token embeddings. These results show that last-token embeddings outperform the alternatives across all three benchmarks, showing the effectiveness of last-token embeddings method.
>
>
> | Method | StrategyQA    | GSM8K    | MMLU    | Avg.    |
> |----------------|------|------|------|------|
> | LASeR (first)  | 83.59 | 73.03 | 67.03 | 74.55 |
> | LASeR (avg.)  | 83.22 | 73.47|  66.52 | 74.40 |
> | LASeR (last) | **85.96** | **74.75** | **68.24** | **76.32** |
>
> ---
>
> > 2. …only one RM is selected per batch, but batches typically contain diverse samples with varying difficulties and type of problems. Consequently, averaging embeddings from multiple samples could produce a noisy estimate of context. Could the authors clarify or provide insights into why their method succeeds despite this issue?
>
> As noted in footnote 2 of the submission, we experimented with different batch sizes to evaluate their impact. Using a batch size of 1 yielded comparable performance to a batch size of 16 but was significantly less efficient in training the LLM due to the increased computational overhead. Based on this, we opted to use a batch size of 16 for a better trade-off between performance and efficiency. We will include this discussion in the main text in our final version.
>
> For datasets like Wildchat, which contain clearly defined and diverse categories, we structure the batches such that each batch consists of data belonging to a single category. This setup minimizes the risk of mismatches between the RM and the batch data, as each RM is evaluated on its most relevant data category. For reasoning datasets where such predefined categories do not exist, shuffling the data during training ensures diverse data and good training signal for LLM within each batch. Even if individual batches vary, the repeated exposure to diverse inputs allows the bandit to learn which RMs are more helpful overall. Over training iterations, this appears to provide a robust signal for the bandit to learn effective RM selection, as demonstrated by LASeR’s consistent gains across datasets and the convergence behavior of MAB rewards shown in Appendix B, Fig. 9.
>
> ---
>
> > 3. How does the accumulated selection frequency for each RM behave—does it change abruptly, or is there a gradual transition between RMs as certain abilities are learned?
>
> We would like to point to an ablation study in which we demonstrated that LASeR selected RM adjusts to the query in Appendix B (lines 1157-1176, Fig. 6 and Fig. 7). These figures show that LASeR selects different RMs depending on the task type. For example, on WildChat, creative prompts favor Olmo and Eurus, while math prompts are dominated by Qwen. Similarly, on LongBench, Zephyr is strongly preferred for summarization and few-shot learning, while Qwen is favored for multi-document QA. This task-aware behavior indicates that LASeR automatically learns to associate RM effectiveness with query type, validating the contextual adaptation capability of our approach.
>
> In addition to these per-task dynamics, we showed in Appendix B (lines 1183-1189, Fig. 9) that the normalized MAB rewards stabilize after 6-8 training iterations. RMs like Qwen and Zephyr, which are more informative for reasoning tasks, retain higher average rewards, indicating that the bandit consistently identifies and prioritizes stronger RMs over time.
>
> ---
>
> > 4. Additionally, could you perform an ablation study on the parameter \alpha. For example, setting \alpha represents greedy selection; does introducing exploration (\alpha > 0) significantly impact performance?
>
> Based on this suggestion, we conduct an ablation study to assess the impact of the exploration parameter $\alpha$ in the LinUCB algorithm. The results show that for reasoning datasets, disabling exploration $\alpha$ leads to suboptimal performance, particularly in early training stages where the bandit may overfit to a suboptimal RM. In contrast, enabling exploration ($\alpha > 0$) improves performance by allowing the model to discover and leverage more informative RMs over time. We find that setting $\alpha$ in the range of 0.4 to 0.6 consistently yields the best results across tasks, offering a good balance between exploration and exploitation. This is consistent with our observations on the performance on dev set for reasoning tasks.
>
> | $\alpha$ (Accuracy)     | 0.0   | 0.2   | 0.4   | 0.6   | 0.8   | 1.0   |
> |------------|-------|-------|-------|-------|-------|-------|
> | StrategyQA | 83.11 | 85.29 | 85.96 | 85.32 | 84.53 | 83.77 |
> | GSM8K      | 71.68 | 73.22 | 74.10 | 74.75 | 74.63 | 73.94 |
>
> We hope our response has addressed all of your questions and will allow you to revisit your score.
>
> ---
>
> **References**
>
> Wang, Liang, et al. "Improving text embeddings with large language models." ACL 2024.
>
> BehnamGhader, Parishad, et al. "Llm2vec: Large language models are secretly powerful text encoders." COLM 2024.

---

> > ### Author Response · Authors · 2025-08-05
> >
> > Thank you once again for your valuable feedback on our paper. Since the author–reviewer discussion period is drawing to a close, with only two days left before the August 6 deadline, we wanted to check in to see whether our response, which included clarification on how LASeR uses last-token embeddings and ablation studies showing LASeR performs adaptive and consistent RM selection throughout training, has addressed all your questions and will allow you to revisit your score. Otherwise we would be happy to engage further and address any further questions you might have in the remaining few days of the discussion period.

---

> > ### Comment · Reviewer_3mJq · 2025-08-09
> >
> > Thank you to the authors for addressing my concerns. I have increased my score to 4 in recognition of the extensive experimental design and the thorough responses provided. However, I have not rated it higher because I still believe the proposed approach is not the most effective for addressing the problem, and the novelty remains somewhat limited.

---

### Official Review · Reviewer_YWzS · 2025-07-06

**Clarity:** 3
**Significance:** 3
**Originality:** 3
**Rating:** 4
**Confidence:** 3

**Summary:**

This paper aims to address a key limitation in LLM RL training: using one fixed reward model may not generalize well across different tasks. The authors introduced LASeR, a method that automatically selects the most suitable reward model for each training example using multi-armed bandit algorithms. More specifically, each reward model is treated as an “arm” in the bandit, and the algorithm dynamically selects which reward model to use for each batch of training examples based on contextual information and past performance. The "reward" for the bandit is the negative training loss, as an indicator for how well the LLM learned from the preference data generated by that reward model. The author reports consistent improvements for commonsense and math reasoning tasks, as well as open-ended instruction-following tasks.

**Questions:**

* Would it be possible to incorporate binary rewards into the LASeR framework? Binary rewards (i.e., whether the final answer is correct or not) tend to work very well in domains like math or code, and have been reported to perform better than learned reward models while being more compute-efficient.
* While using the negative training loss as the "reward" for the bandit makes intuitive sense, have you explored other learning signals? How do they compare?

**Ethical Concerns:**

["NO or VERY MINOR ethics concerns only"]

**Final Justification:**

Authors' rebuttal enhances the empirical contribution, I increased my quality and significance score.

**Limitations:**

yes

**Quality:**

3

**Strengths And Weaknesses:**

### Strengths:
* The paper is well motivated and addresses an important issue in RL training, the selection of reward models.
* Framing reward model selection as a multi-armed bandit problem is elegant and novel.
* The author conducted comprehensive experiments across multiple models and tasks, and observed meaningful improvement.
* The paper is well-written , well-structured and easy to follow.

### Weakness:
* While the paper addresses an important problem—how to dynamically select reward models for different tasks—for math problems where the final answer can be easily verified with a rule-based verifier, a common approach in recent literature is to use binary rewards in RL. This method has been reported to outperform reward models and should be included as an important baseline for the math tasks.
* LASeR introduces additional compute overhead during LLM training, and the learned multi-armed bandits are limited to a pre-selected set of reward models. What if I want to add or update a reward model? Do I have to retrain the MAB?

---

> ### Author Rebuttal · Authors · 2025-07-30
>
> We thank you for your comments and suggestions, and we are glad that you found our method to be “well motivated” and “novel”, and our results to represent a “meaningful improvement.” Below, we provide specific responses to each of your points.
>
> ---
>
> > While the paper addresses an important problem…a common approach in recent literature is to use binary rewards in RL. This method has been reported to outperform reward models and should be included as an important baseline for the math tasks.
>
> We thank the reviewer for this suggestion. We implement a Correctness Reward baseline for the reasoning datasets. Specifically, we assign a reward of 1 if the generated answer matches the correct final answer and 0 otherwise. We maintain the same iterative training setup as in our main experiments, sampling $n= 30$ responses and $P = 10$ pairs of (correct, incorrect) per query. We report the results in the table below. These results show that **while the correctness-based reward performs competitively, especially on GSM8K (74.56% vs. 74.75% accuracy), LASeR outperforms it across all tasks, including commonsense reasoning tasks (StrategyQA and MMLU)**.
>
> This is likely because correctness-based rewards are well-suited for tasks with clearly verifiable outputs, such as math or code, but they struggle to capture the multi-step reasoning and contextual understanding needed in **factual or open-domain QA, where answers are often ambiguous or depend on background knowledge**. In these cases, other forms of supervision such as process-based rewards (Ma et al., 2025) or human preference signals modeled by reward models (Peng et al., 2025) are typically more effective. LASeR is designed to leverage this by selecting among multiple RMs, allowing it to perform well across a variety of task types. As we show below, LASeR can also integrate correctness-based rewards, combining their strengths with those of preference-based models.
>
> | Method | StrategyQA    | GSM8K    | MMLU    | Avg.    |
> |----------------|------|------|------|------|
> | Correctness Reward  | 83.42 | 74.56 |  66.35 | 74.78 |
> | LASeR | **85.96** | **74.75**| **68.24** | **76.32** |
>
> ---
>
> > Would it be possible to incorporate binary rewards into the LASeR framework?
>
> **Yes, binary rewards can be incorporated into the LASeR framework**. In our setup, each RM is separated, so we can simply define a new RM that assigns a reward of 1 to correct final answers and 0 to incorrect ones, and include it in the MAB’s candidate set.
>
> **Setup.** To test this, we run an additional experiment on GSM8K, where we add the correctness-based RM to the existing pool of learned RMs (Olmo, Eurus, Qwen, Zephyr). To further test LASeR’s ability to adaptively select appropriate RMs, we create a mixed dataset combining GSM8K with WildChat (subsampled to 1.2K examples per category to ensure balance). Since correctness-based signals are not applicable to WildChat’s open-ended prompts, we assign a random binary reward (0 or 1) for those examples to simulate a misleading RM.
>
> **Results.** LASeR achieves an accuracy of 75.06% accuracy on the GSM8K test set, improving over LASeR without correctness rewards by 0.31%. We also report the utilization rate (%) of each RM in the pool (Olmo, Eurus, Qwen, Zephyr, correctness/random) for two types of queries: GSM8K and WildChat in the table below. On GSM8K, LASeR heavily leverages the correctness-based RM (40.26%) alongside strong learned RMs like Qwen (43.58%), indicating that **it effectively identifies and uses the most reliable supervision**. In contrast, for WildChat, LASeR downweights the misleading correctness-based RM (2.84%) and shifts toward more suitable RMs like Zephyr and Qwen. These results demonstrate LASeR’s ability to **adapt RM usage based on task domain, even in the presence of noisy or misleading supervision**. This is consistent with our results on the robustness of LASeR to noisy rewards (Section 6, line 426 and Appendix C, lines 1295-1306).
>
> |  Utilization Rate (%)            | Olmo | Eurus | Qwen  | Zephyr | Correctness/Random |
> |--------------|------|-------|-------|--------|----------------|
> | GSM8K        | 4.02 | 2.32  | 43.58 | 19.82  | 40.26          |
> | WildChat      | 9.09 | 16.41 | 27.75 | 45.90  | 2.84           |
>
> ---
>
> > LASeR introduces additional compute overhead during LLM training, and the learned multi-armed bandits are limited to a pre-selected set of reward models. What if I want to add or update a reward model? Do I have to retrain the MAB?
>
> Although LASeR introduces additional compute overhead during LLM training, this is **minimal compared to iterative DPO training**. We showed in lines 1042-1047 that LASeR only adds 1.27 seconds to the training pipeline per batch. In comparison, a forward and backward pass of the LLM takes 36.39 seconds on 4 RTX A6000-48G GPUs.
>
> **Adding or updating a reward model in LASeR does not require retraining the entire MAB from scratch**. LinUCB maintains independent statistics ($A_k$, $B_k$) for each RM arm. To add a new RM, we can initialize its parameters and begin exploration, allowing the bandit to learn its utility online. When adding a new RM to LASeR, only the new RM's bandit parameters need to be initialized (e.g., $A_k=I$, $b_k=0$). The existing RMs' statistics remain unchanged, meaning the MAB retains all prior learning about their utility. This allows the bandit to continue exploiting well-performing existing RMs while gradually exploring the new RM. Similarly, if an existing RM is updated (e.g., improved or retrained), the MAB can re-learn its statistics gradually through continued training.
>
> To empirically demonstrate this, we conduct an experiment where we resume training Llama-3-8B from a saved checkpoint and add a new, stronger RM nicolinho/QRM-Llama3-8B  based on the latest RewardBench leaderboard to the current pool of RMs. We observe that the MAB quickly adapts to the new RM, with training converging after just 2 additional iterations and achieving 75.58% accuracy on the GSM8K test set, improving over LASeR without QRM-Llama3-8B by 0.83%. We also run an additional experiment where we reset the MAB parameters for an existing RM (Qwen) and then resume training Llama-3-8B from a saved checkpoint. Training converges after 3 additional iterations and achieves comparable accuracy to the original checkpoint before the reset (74.72% vs. 74.75% accuracy).
>
> Besides accuracy, we also show the utilization rates (%) of each RM before (Checkpoint), after (Transfer) adding the new RM, and restart (Restart) in the table below. These results show that LASeR can **effectively integrate new RMs and adapt its selection strategy, shifting toward using stronger RMs when they offer more informative supervision, while still retaining and utilizing previously effective RMs** like Qwen for reasoning tasks. Additionally, in our restart experiment, where we reset the MAB parameters of Qwen and resume training, demonstrates that LASeR can quickly **relearn the utility of a strong RM**, returning to similar usage levels with just a few additional iterations.
>
> | Utilization Rate (%)              | Olmo | Eurus | Qwen  | Zephyr | QRM   |
> |--------------|------|-------|-------|--------|--------|
> | Checkpoint   | 6.26 | 2.06  | *66.81* | 24.87  | -   |
> | Transfer (Add QRM)    | 1.68 | 0.77  | 35.08 | 8.93   | *53.54*  |
> | Restart (Qwen)  | 7.68 | 2.47 | *65.25* | 24.59 | -   |
>
> ---
>
> > While using the negative training loss as the "reward" for the bandit makes intuitive sense, have you explored other learning signals? How do they compare?
>
> We empirically compared our approach (using negative training loss as the MAB reward) with a variant that uses accuracy difference ($\text{Acc}(y_w) - \text{Acc}(y_l)$), which relies on ground-truth answers. As noted in Section 3, lines 148-149 and shown in Appendix C (lines 1280-1283, Table 13), using the negative training loss results in stronger performance gains, suggesting it serves as a reliable proxy for RM signal quality. The intuition is that DPO loss directly captures how clearly the model distinguishes between preferred and rejected responses, helping the model to sharpen preference rankings and increase confidence in distinguishing between chosen and rejected responses (see lines 174-187 in our submission).
>
> We hope our response has addressed all of your questions and will allow you to revisit your score.
>
> ---
>
> **References**
>
> Ma, Ruotian, et al. "S $^ 2$ R: Teaching LLMs to Self-verify and Self-correct via Reinforcement Learning." ACL 2025.
>
> Peng, Hao, et al. "Agentic reward modeling: Integrating human preferences with verifiable correctness signals for reliable reward systems." ACL 2025.

---

> > ### Author Response · Authors · 2025-08-05
> >
> > Thank you once again for your valuable feedback on our paper. Since the author–reviewer discussion period is drawing to a close, with only two days left before the August 6 deadline, we wanted to check in to see whether our response, which included new results for a correctness-based reward baseline and new experiments demonstrating how binary rewards (or any new RMs) can be efficiently incorporated to LASeR, has addressed all your questions and will allow you to revisit your score. Otherwise we would be happy to engage further and address any further questions you might have in the remaining few days of the discussion period.

---

### Note · Authors · 2025-08-12

Dear Reviewers, ACs, and SACs,

Thank you for taking the time to review our paper. In our rebuttal and during the discussion period, we tried our best to clarify and address the reviewers’ concerns, and we would like to take the final remark to summarize them below for your consideration.

**Novelty and Contribution.** As Reviewer 3mJq and Reviewer PKfF raised questions about the novelty of our paper, we would like to emphasize again that, while contextual MAB is well-established, applying it to preference-based LLM training over multiple RMs is **novel** and **not a direct extension**. **LASeR addresses limitations** in both **standard iterative LLM training pipelines** that rely on a single RM, which risk overfitting to one RM (prone to unreliable rankings and reward hacking) and cannot generalize to other domains, and **RM ensemble methods**, which incur higher compute costs and may suffer from conflicting signals between RMs.

Compared to **standard MAB settings**, our setting introduces unique challenges: (1) dynamic reward distributions as RMs co-evolve with the LLM during training, (2) lack of explicit supervision on which RM is optimal per query, addressed via a justified proxy reward from negative DPO loss, and (3) online adaptation to new, noisy, or domain-specific RMs without retraining.

We believe our paper presents a novel, practical, and extensible LASeR framework and demonstrates strong empirical results across diverse tasks (reasoning, instruction-following, and long-context understanding), which we believe are in line with NeurIPS originality guidelines.

**Additional Experiments.** To address reviewer comments, we added multiple experiments to strengthen our work:
* A correctness-based reward baseline and mixed-task experiments showing LASeR’s adaptive RM integration to a new correctness-based RM or an updated RM (Reviewer YWzS).
* Ablations on sentence embedding choice, RM selection dynamics, and exploration parameter $\alpha$, plus batching method clarification (Reviewer 3mJq).
* Hypothesis tests showing statistically significant gains (Reviewer PKfF).

We are happy that **Reviewer 3mJq and Reviewer PKfF acknowledged the extensive new experiments** we provided. Reviewer 3mJq also increased their score in light of the additional clarifications and ablations in our rebuttal.

Thank you again for your time and consideration. We will incorporate all feedback into the final version of our paper.

Best regards,

Authors of Submission 24282

---

### Decision · Program_Chairs · 2025-09-17

**Decision:**

Accept (poster)

**Comment:**

This paper introduces LASeR, a novel and practical framework that uses a MAB approach to adaptively select the most suitable reward model for each instance during LLM alignment. The reviewers generally agree that the paper to be well-written, well-motivated, and supported by comprehensive experiments. Initial concerns regarding the method's novelty, the statistical significance of the results, and specific implementation details were mostly addressed in the rebuttal. These make it a valuable addition to the field. Therefore, I recommend acceptance.